# Comparison of Body Characteristics, Carotenoid Composition, and Nutritional Quality of Chinese Mitten Crab (*Eriocheir sinensis*) with Different Hepatopancreas Redness

**DOI:** 10.3390/foods13070993

**Published:** 2024-03-25

**Authors:** Honghui Guo, Jingang Zhang, Yidi Wu, Xiangzhong Luo, Zhiqiang Xu, Jianlin Pan, Guiwei Zou, Hongwei Liang

**Affiliations:** 1Yangtze River Fisheries Research Institute, Chinese Academy of Fishery Sciences, Wuhan 430223, China; guohonghui@yfi.ac.cn (H.G.); zhangjingang@yfi.ac.cn (J.Z.); wuyd@yfi.ac.cn (Y.W.); lxz@yfi.ac.cn (X.L.); zougw@yfi.ac.cn (G.Z.); 2College of Fisheries and Life Science, Shanghai Ocean University, Shanghai 201306, China; 3Key Laboratory of Freshwater Crustacean Genetic Breeding and Cultivation (Co-Construction by Ministry and Province) Ministry of Agriculture and Rural Affairs, Freshwater Fisheries Research Institute of Jiangsu Province, Nanjing 210017, China; zhiqiangx@163.com (Z.X.); jianlinpan2006@126.com (J.P.)

**Keywords:** hepatopancreas chroma, carotenoids, paraffin section, fatty acids, amino acids, DHA, EPA, nutritional quality, *Eriocheir sinensis*

## Abstract

In this study, we investigated the body characteristics, carotenoid composition, and nutritional quality of *Eriocheir sinensis* with different hepatopancreas redness (*a**). We distributed the crabs into two groups based on the hepatopancreas *a** values and compared their body characteristics, chroma, carotenoid composition, and protein, lipid, total sugar, amino acid, and fatty acid content via paired *t*-test. The results revealed that the relationships between hepatopancreas *a** values and crab quality are sex specific. In female crabs, the differences in nutritional characteristics were evident mainly in the hepatopancreases and ovaries. In the redder hepatopancreases, the content of zeaxanthin and β-carotene increased, and the levels of C22:6n3 and C20:5n3 decreased (*p* < 0.05). In the ovaries, the content of astaxanthin, canthaxanthin, β-carotene, umami, and sweet amino acids were lower in the redder hepatopancreas crabs (*p* < 0.05). In male crabs, there were positive relationships between hepatopancreas *a** and amino acid and fatty acid content. The content of leucine, arginine, and total umami amino acids in muscles and of unsaturated fatty acids and n-6 polyunsaturated fatty acids in hepatopancreases and testicles increased with increasing hepatopancreas *a** values (*p* < 0.05). Therefore, the redder the hepatopancreas, the higher the nutritional quality of male crabs.

## 1. Introduction

Chinese mitten crab (*Eriocheir sinensis*), a native species from the East Pacific coast of China and the Korean Peninsula, is widely distributed in the United States and some European and Asian countries. From the 1930s to the early 21^st^ century, their range expanded to include the North Sea, Baltic Sea, Mediterranean Sea, Spain, Portugal, Eastern Europe, North America, and Iran [1]. Globally, the aquaculture of *E. sinensis* is rapidly expanding, reaching a productivity of 775.9 thousand tons in 2020 [2]. China, the main producer of *E. sinensis*, had a crab yield of 815.32 thousand tons in 2022, and the output value has increased to more than CNY 50 billion [3]. *E. sinensis* is popular among consumers in China and other Asian countries. *E. sinensis* is rich in carotenoids, fatty acids, amino acids, and volatile flavor compounds, such as essential amino acids (EAAs), C20:5n3 (EPA), and C22:6n3 (DHA) [4,5]. As a result, *E. sinensis* has emerged as a favored culinary delight. Recently, the relationship between the chroma traits and nutritional quality of *E. sinensis* has been researched to develop functional foods and to breed novel crab varieties.

The chroma of crustaceans, the most important sensory indicator of quality, affects sales, prices, and consumer acceptability [6]. *Penaeus monodon* is assessed for color using a subjective color grading chart, with scores ranging from 1 (low pigmentation) to 12 (high pigmentation). Highly colored prawns have higher prices as consumers prefer orange and intense orange cooked shrimp [7]. Carotenoids and crustacyanin play important roles in the chroma of crabs. Carotenoids cannot be synthesized de novo in crabs; therefore, *E. sinensis* chroma is significantly associated with dietary factors and environmental conditions [8,9]. Background color and light intensity are highly effective at redistributing carotenoid pigments, to make shrimp both darker or lighter [10]. Recently, associations between the carapace color and the carotenoid composition and nutritional quality of *E. sinensis* have been clarified. A higher content of total astaxanthin and astaxanthin ester in the hepatopancreases and ovaries was observed in purple crabs compared to green-black crabs [8]. Compared with green-black crabs, purple crabs have a higher content of EPA and C20:4 (ARA) in females and males, respectively [8]. In white crabs, the proportion of C20:4n6 in the muscles is higher than in green-black crabs, but the proportion of C18:0, C18:2n6, and DHA/EPA is lower in the hepatopancreases [11]. In addition, a higher content of crude protein in the gonads and of moisture in the muscles was detected in crabs with higher *a** and *b** values in freeze-dried carapaces [12]. In contrast, purple crabs are considered to be of high quality, with more carotenoids and greater fatty acid content in the ovaries and the hepatopancreases. The chroma of the carapace has a significant correlation with the nutritional quality of crabs. In general, the color of the hepatopancreas is more important to consumers than carapace color as it is the main edible part of crabs.

Studies have shown that the nutritional quality of crabs decreases as the hepatopancreas color becomes lighter. Hepatopancreatic necrosis disease (“shuibiezi” in Chinese) and white hepatopancreas syndrome have been increasingly observed in *E. sinensis*, which causes the hepatopancreas color to change from golden yellow to yellow-white and then to white [13]. In males, the content of amino acids in the muscles and gonads and of monounsaturated fatty acids (MUFAs) in the hepatopancreases was lower in white hepatopancreas crabs than in yellow hepatopancreas crabs [14]. In females, the hepatopancreas content of polyunsaturated fatty acids (PUFAs) and n-6 PUFAs was lower in white and yellow-white hepatopancreas crabs than in yellow hepatopancreas crabs [15]. Additionally, the content of total bile acids, bitter-free amino acids, and lutein was significantly higher in brown hepatopancreas crabs than in orange hepatopancreas crabs, while β-carotene was significantly lower in brown hepatopancreas than in orange hepatopancreas crabs [16]. Previous studies have compared the quality of crabs in correlation with hepatopancreas colors between healthy and diseased crabs. The findings revealed that the color of the hepatopancreas could be used to characterize disease progression. In healthy crabs, the hepatopancreas color has revealed different *a** (redness) values. Even though there is no standard color for *E. sinensis*, it is traditionally assumed that a “redder” hepatopancreas indicates superior quality due to its association with freshness and carotenoids. Crabs are important sources of natural carotenoids for other animals, including humans. Carotenoids play important potential roles in human health by acting as biological antioxidants, protecting cells and tissues from the damaging effects of free radicals and singlet oxygen, and inhibiting the development of certain cancers. Generally, hepatopancreas redness is judged by the chroma of the cuirass and abdomen of *E. sinensis* (Figure 1). The redness and brightness of the hepatopancreas are appealing to consumers. However, the different nutritional profiles and flavors of *E. sinensis* with different hepatopancreas *a** values have not been comprehensively clarified. Significant increases in carotenoid and fatty acid content were obtained with increasing hepatopancreas *a** values in *E. sinensis* fed astaxanthin [6]. Compared with green-black crabs, the EPA content in muscles, hepatopancreases, and ovaries were higher in female purple crabs with higher hepatopancreas *a** values [8]. Consequently, the carotenoid composition and nutritional quality of *E. sinensis* might be positively correlated with hepatopancreas *a** values.

The aim of this study was to clarify the qualities of *E. sinensis* with different hepatopancreas redness. Jiangsu is the main producing area of *E. sinensis* in China. The yield of crabs in Jiangsu was 37.41 thousand tons accounting for 45.88% of the entire production in 2022 [3]. Therefore, in this study, *E. sinensis* was sampled from Zhenjiang city (Jiangsu province). The crabs were distributed into light and dark color groups according to hepatopancreas *a** values. The body characteristics, chroma (cuirass, abdomen, hepatopancreas, and gonad), histology, carotenoids, protein, lipid, total sugar, amino acids, and fatty acids were measured. The results of the present study not only facilitate the quality assessment of *E. sinensis,* but also provide useful information on different redness values of hepatopancreas of *E. sinensis*.

## 2. Materials and Methods

### 2.1. Sample Collection and Preparation

We collected *E. sinensis* from Zhenjiang, Jiangsu Province, China, in November 2022 when the crabs became sexually mature. Among 540 *E. sinensis* (296 female crabs and 244 male crabs) from one pond (119°70′76.74″ E, 31°94′82.17″ N), the average hepatopancreas *a** values in females and males were 23 and 24, respectively. The redder hepatopancreas crabs were differentiated through the assessment of color using both a spectrophotometer ("*a**" parameter) and subjective evaluation [7]. Based on the ascending order of hepatopancreas *a**, *E. sinensis* samples were randomly chosen, with the top 20% constituting the light color groups (LCG) and the bottom 20% forming the dark color groups (DCG). This approach ensured significant differences in hepatopancreas *a** values (Table 1) and in visual differences on the chroma of cuirass and abdomen of crabs from different groups (Figure 2 and Figure 3). Thirty-two crabs were selected and distributed into the LCG and DCG (sixteen female crabs: eight crabs in LCG and eight crabs in DCG; sixteen male crabs: eight crabs in LCG and eight crabs in DCG; Table 1 and Figure 2 and Figure 3). Crabs were gently blotted with a towel to remove surface moisture. We assessed the body characteristic indexes and measured the cuirass and abdomen chroma. The muscles, hepatopancreases, and gonads of each crab were dissected at 4–7 °C and weighed immediately. Subsequently, we measured the chroma of the hepatopancreases and gonads from each crab. A small portion of the hepatopancreas and gonad from each crab was fixed in 10% neutral-buffered formalin for paraffin section analysis. The remaining tissues were frozen in liquid nitrogen and stored at −80 °C for subsequent analysis. The animal study was reviewed and approved by the animal care regulations of the Yangtze River Fisheries Research Institute, Chinese Academy of Fishery Sciences (2022-yfi-ghh-03).

The gonad–somatic index (GSI) and hepatopancreas–somatic index (HSI) were calculated using the following equations:GSI = gonad weight (g)/body weight (g) × 100
HSI = hepatopancreas weight (g)/body weight (g) × 100.

### 2.2. Color Parameters Measurement

We measured lightness (*L**), redness (*a**), and yellowness (*b**) of cuirass, abdomen, hepatopancreas, and gonad in a DS-700D spectrophotometer (Hangzhou CHNSpec Technology Co., Ltd., Hangzhou, China). We collected three measurements of each sample surface according to the method reported by Li et al. [8]. *L** corresponds to the degree of lightness where *L** > 0 is white and *L** < 0 is black. The parameters *a** and *b** are two chromatic components, where *a** > 0 when colors are red, *a** < 0 when colors are green, *b** > 0 when colors are yellow, and *b** < 0 when colors are blue.

### 2.3. Hepatopancreas and Gonad Paraffin Section Analysis

The hepatopancreases and gonads were preserved in 10% neutral-buffered formalin for 48 h, embedded in paraffin wax, sectioned (5 µm), and stained with hematoxylin and eosin (H&E). Histopathological assessment was performed using light microscopy (Nikon H600L Microscope and image analysis system, Tokyo, Japan). A detailed protocol is provided in the Appendix A.

### 2.4. Total Sugar, Lipid, and Protein Analysis

We detected total sugar content based on method GB/T 9695.31-2008 [17]. Briefly, samples were weighed, hydrolyzed with hydrochloric acid, filtered, and adjusted to 250 mL with water. We measured the absorbance of the samples at 490 nm. Total sugar content was calculated using a standard curve.

We measured lipid content according to the method reported by Qin et al. [18]. Briefly, samples were digested with hydrochloric acid and mixed with ethanol and anhydrous ether until the upper liquid was clear. Anhydrous ether was recovered and dried to constant weight (the difference between the two weights was ≤2 mg).

Protein content was determined using the Kjeldahl method according to GB 5009.5-2016 [19] and a 6.25 nitrogen-to protein conversion factor.

All detailed protocols are provided in the Appendix A.

### 2.5. Carotenoids Compositions Analysis

Carotenoids were extracted from the hepatopancreases and ovaries according to the method of Li et al. [20]. Briefly, we homogenized samples in acetone in an ultrasonic cleaner for dissociation of the pigment. Subsequently, we removed the homogenates after centrifugation. We re-extracted the samples in the dark using acetone until the samples were colorless. Total carotenoids were analyzed spectrophotometrically at 478 nm against acetone as the blank. The concentration of total carotenoids was estimated based on commercially available standards of β-carotene in acetone. A portion of the carotenoids extracted from each sample was used to analyze free carotenoids. Samples were passed through a 0.45 μm syringe filter for high-performance liquid chromatography (HPLC) analysis. Carotenoids were identified based on five commercially available standards of known concentration (astaxanthin, lutein, zeaxanthin, and β-carotene from Shanghai Yuanye Bio-Technology Co., Ltd., Shanghai, China; canthaxanthin from Dr. Ehrenstorfer GmbH., Augsburg, Germany). We performed quantitative determination using an external standard method. A detailed protocol is provided in the Appendix A.

### 2.6. Amino Acids and Fatty Acids Analysis

Amino acid content was measured according to the method reported by Zhang et al. [21]. Samples were weighed and homogenized in TCA. The supernatant was collected after ultrasonication. The pH of the supernatant was adjusted to 2.0, and the solution was passed through a 0.22 µm syringe filter for quantitative analysis using an amino acid analyzer (L-8800; Hitachi Co., Ltd., Tokyo, Japan). We separated the samples using an Inertsil ODS-3 C18 column (4.6 mm × 150 mm, 7 μm; GL Sciences Inc., Tokyo, Japan). Detailed protocols are provided in the Appendix A.

The peak area percentage method was carried out to measure fatty acids [22]. Crude lipids were further processed for fatty acid analysis. We performed saponification using potassium hydroxide. After cooling to room temperature, we added sulfuric acid for further transesterification. The upper organic layer was diluted with n-hexane after centrifugation, and a 0.22 μm filtration membrane was used to remove impurities. FAMEs were determined using a gas chromatograph (Agilent-2890A; Agilent Technologies Co., Ltd., Santa Clara, CA, USA), equipped with a flame ionization detector. Separation was conducted using a capillary column (30 m, 0.25 mm, 0.25 μm; DB-WAX; Agilent Technologies Co., Ltd., Beijing, China) with split injection (10:1) and helium (>99.99%) at a constant flow of 0.8 mL/min. Identification was accomplished by comparing the retention times of samples with those of standards, and the results were expressed as the relative weight percentage of the identified fatty acids based on the peak areas obtained using the software from the instrument. Detailed protocols are provided in the Appendix A.

### 2.7. Statistical Analysis

Statistical analysis was performed using SPSS 22.0 for Windows (SPSS Inc., Armonk, NY, USA). The data were expressed as mean ± standard deviation. To compare LCG and DCG, we used an independent sample *t*-test. Differences were measured and considered to be statistically different at *p* < 0.05.

## 3. Results

### 3.1. Differences in Body Characteristics, Hepatopancreas, and Gonad Paraffin Section in Crabs with Different Hepatopancreas Redness

In female crabs, there was a significantly visual difference in the chroma of the cuirass and abdomen of crabs from different groups (Figure 2). The values of hepatopancreas *a** and *b** were significantly higher in DCG than in LCG (Table 1; *p* = 0.000 and *p* = 0.010, respectively). The values of cuirass and abdomen *a** increased significantly in redder hepatopancreas crabs (*p* = 0.039 and *p* = 0.009, respectively) due to the fact that the inner sides of the cuirass and abdomen were filled with hepatopancreas. Cuirass *a** values were positively correlated with hepatopancreas *a** values (Appendix A; *r* = 0.67; *p* = 0.004). Therefore, the hepatopancreas chroma can be assessed through the cuirass chroma. There were no significant differences in hepatopancreas and ovarian paraffin sections between DCG and LCG (Figure 4). The hepatopancreases showed the normal hepatic tubules, lucid lumen, columnar hepatocytes, round nuclei at the base, and complete basal membrane structure (Figure 4). The ovaries presented several round mature oocytes (stage V). The yolk granule was evenly distributed in the cytoplasm of mature oocytes. The paraffin sections showed that the crabs were healthy and sexually mature. However, there were no significant differences in HSI, GSI, and body characteristics between the two groups (*p* > 0.05).

In male crabs, there was a significantly visual difference in the chroma of the cuirass and abdomen of crabs from different groups (Figure 3). The hepatopancreas and abdomen *a** values increased significantly in redder hepatopancreas crabs (Table 1; *p* = 0.001 and *p* = 0.037, respectively). We obtained a significant positive correlation between hepatopancreas *a** and abdomen *a** values (Appendix A; *r* = 0.50; *p* = 0.049), which indicated that hepatopancreas chroma could be assessed through the abdomen chroma. Additionally, HSI increased significantly with increasing hepatopancreas *a** values (*p* = 0.028), which revealed that male crabs with a redder hepatopancreas might have a greater nutritional quality. There were no significant differences in the hepatopancreas and testicle paraffin sections between LCG and DCG (Figure 5). The hepatopancreases presented normal hepatic tubules, lucid lumen, columnar hepatocytes, round nuclei at the base, and a complete basal membrane structure. The testes, at the mature stage, presented large vas deferens volume and coarse tubular structure with main cell types of sperm cells and sperms. These results indicate that the crabs were healthy and sexually mature.

### 3.2. Differences in Carotenoids and Biochemical Compositions in Hepatopancreas and Gonad of Crabs with Different Hepatopancreas Redness

In female crabs, we detected significant differences in the hepatopancreas total carotenoids (*p* = 0.000), zeaxanthin (*p* = 0.038), and β-carotene (*p* = 0.000) between the two groups. β-carotene and zeaxanthin were the main carotenoids contributing to color. In the ovaries, the content of astaxanthin (*p* = 0.040), canthaxanthin (*p* = 0.050), and β-carotene (*p* = 0.011) decreased significantly with increasing hepatopancreas *a** values (Figure 6). Total sugar increased significantly in the hepatopancreases (*p* = 0.022) and ovaries (*p* = 0.015) with increasing hepatopancreas *a** values. In male crabs, the levels of carotenoids increased in redder hepatopancreas crabs; however, the results were not significant (*p* > 0.05). The hepatopancreas protein content increased significantly with increasing hepatopancreas *a** values (Figure 6; *p* = 0.042), which revealed that male crabs with a redder hepatopancreas had a greater nutritional quality. Consequently, the hepatopancreases of *E. sinensis* represent a good source of carotenoids, especially the redder hepatopancreases.

### 3.3. Differences in the Compositions of Amino and Fatty Acids in Crabs with Different Hepatopancreas Redness

In female crabs, there were no significant differences in the amino acid content of hepatopancreases and muscles between the two groups (Table 2; *p* > 0.05). The content of ovarian Asp, Thr, Ser, Glu, Gly, Val, Pro, total umami amino acids (TUAAs), and total sweet amino acids (TSAAs) decreased significantly with increasing hepatopancreas *a** values (*p* < 0.05), which indicated that the redder hepatopancreas crabs had a lower nutrient value in ovaries with lower levels of amino acids. With respect to fatty acids, the content of MUFAs, C18:1n9c, C20:2, C20:4n6, EPA, and DHA decreased in redder hepatopancreases (*p* < 0.05). The content of C16:1 and C18:3n3 in muscles increased with increasing hepatopancreas *a** values (*p* < 0.05). Similarly, C18:3n3 and C20:3n3 in ovaries increased with increasing hepatopancreas *a** values (*p* > 0.010 and *p* > 0.030, respectively); however, EPA content decreased in ovaries (*p* = 0.034). Therefore, the differences in nutritional characteristics were evident in the hepatopancreases and ovaries of female crabs with different hepatopancreas *a** values.

In male crabs, there was no significant difference in the hepatopancreas amino acid content between the two groups (Table 2; *p* > 0.05). In the testicles, only His content increased significantly with increasing hepatopancreas *a** values (*p* = 0.044). Similarly, the content of Leu, Arg, and TUAAs in muscles increased significantly with increasing hepatopancreas *a** values (Table 2; *p* = 0.040, *p* = 0.039, and *p* = 0.048, respectively). Our findings indicate that redder hepatopancreas crabs had greater nutritional quality and umami muscles. Table 3 shows the fatty acid content in the hepatopancreases of male crabs with different hepatopancreas *a** values. The content of C14:1, C18:1n9c, C18:2n6c, C18:3n3, SFAs, MUFAs, PUFAs, UFAs, n-6 MUFAs, and total fatty acids increased significantly in redder hepatopancreases (*p* < 0.05). Similarly, C16:1, C18:2n6c, C18:3n3, n-6 MUFA, and UFA content increased significantly in testicles (*p* < 0.050) with increasing hepatopancreas *a** values. C18:3n3 increased significantly in muscles with increasing hepatopancreas *a** values (*p* = 0.041). Therefore, the redder the hepatopancreas, the greater the nutritional quality of male *E. sinensis*.

## 4. Discussion

Conventionally, the chroma of cuirass and abdomen of *E. sinensis* has been considered a quick means of assessing hepatopancreas chroma. Our study findings revealed that the *a** values of the cuirass and abdomen increased significantly in redder hepatopancreas crabs. Spearman correlation analysis showed that hepatopancreas *a** values were positively correlated with cuirass and abdomen *a** values in females and males, respectively. These differences might be attributed to the different structures of their cuirass and abdomen (Figure 1). Therefore, the hepatopancreas chroma of mature female and male crabs should be assessed through the chroma of cuirass and abdomen, respectively. However, these relationships need be clarified in immature crabs. Additionally, Li et al. reported different hepatopancreas chroma among crabs with different carapace chroma [23]. In general, chemical analysis and color scoring are the main methods that allow us to characterize the degree of pigmentation in crustaceans. Currently, color scoring is widely applied for the color assessment of aquatic animals such as crabs and shrimps. Color scoring is achieved either visually or spectrophotometrically, which results in low accuracy due to perceived color changes depending on the amount of water and the environment. Therefore, chemical analysis for the degree of pigmentation should be applied to verify our results. Crustacean chroma is the major factor responsible for customer acceptability, because redder hepatopancreases are recognized for their superior nutritional quality [6,24,25]. However, there is no standard color for *E. sinensis*. It is traditionally assumed that redder hepatopancreases are of superior quality. Therefore, consumer studies and market analysis data are required to solidify consumer behavior and preference claims. Previous studies have reported associations between hepatopancreas yellowness and the nutritional quality of *E. sinensis* [14,15]. However, few studies have reported the nutritional quality of crabs with different levels of hepatopancreas redness.

Carotenoids are the basic pigments in crustacean tissues such as the shells, ovaries, and hepatopancreases. In this study, the content of total carotenoids, lutein, zeaxanthin, canthaxanthin, and β-carotene was higher in the crabs with redder hepatopancreases. Spearman correlation analysis showed that carotenoid levels had a positive correlation with hepatopancreas *a** values, especially in female crabs. In *Panulirus cygnus,* there was a 2.4-fold difference in total carotenoid content in shell extracts of red lobsters compared with white lobsters [26]. Similarly, redness (*a**) values, total astaxanthin, and astaxanthin ester content of the carapaces, hepatopancreases, and ovaries was significantly higher in purple crabs than in green-black crabs [8]. β-carotene content was significantly lower in the brown hepatopancreas than in the orange hepatopancreas [16]. Therefore, the scoring of hepatopancreas redness could be evaluated by measuring carotenoid content. Carotenoids play important roles in the physiology of crabs including growth, hepatopancreas nutritional metabolism, molting, and reproduction. Our results showed that the *E. sinensis* body weight increased with increasing hepatopancreas *a** values. In females, a lower content of ovarian carotenoids was observed in crabs with a redder hepatopancreas. The hepatopancreas, the nutrient storage and metabolic center of *E. sinensis*, provides nutrients and non-nutrients for gonad development [27]. The stored carotenoids in the hepatopancreas may be transported to the ovaries through the hemolymph for ovarian development [28]. Therefore, more hepatopancreatic carotenoids might be transported for ovarian developing in female crabs with a lighter red hepatopancreas [29,30]. Free astaxanthin contributes to the yellow, orange, and red in tissues, and crustacyanin-bound astaxanthin contributes to the green and brown [31]. Therefore, different ovarian chroma might be attributed to the synergistic effects of free and esterified astaxanthin levels because esterified astaxanthin cannot bind to crustacyanin. The content and type of carotenoids in crabs are affected by diet and the environment [29]. The pond culture enhanced the levels of protein, amino acids, and specific organic acid derivatives and reduced the levels of peptides and PUFAs compared with crabs in lakes [32]. The content of carotenoids, DHA, EPA, EAAs, and FAAs significantly improved in the male hepatopancreases and muscles under saline water and alkaline water [12]. Therefore, hepatopancreas redness as an indicator of nutritional quality should be researched under different culture conditions. Moreover, genetics should be considered in future studies, because these crabs were sampled from the same environmental and dietary conditions. However, the color parameters (*L**, *a**, *b**) of the carapaces, hepatopancreases, and ovaries have very low heritability [23]. Future studies should validate our findings. Therefore, *E. sinensis* represents a good source of carotenoids, especially redder hepatopancreases. Redder hepatopancreas crabs are optimal for individuals with vitamin A deficiency.

Previous studies have reported a potential association between the hepatopancreas chroma and nutrient composition [8,16]. In female crabs, the hepatopancreases and ovaries had higher levels of total sugars in redder hepatopancreas crabs, which may impart sweetness. However, the free amino acids in hepatopancreases, ovaries, and muscles were lower in the female crabs with a redder hepatopancreas, especially ovarian Thr and Val. These results are consistent with the carotenoid content in the ovaries and indicate a reduction in carotenoids, possibly due to a reduction in amino acids. Dietary astaxanthin may regulate the amino acid metabolism of the hepatopancreas in *Exopalaemon carinicauda* [33]. In fish, Carophyll^®^ pink increased the level of umami amino acids (Asp and Glu) [34]. However, further research is required to determine whether carotenoids are the direct reasons for amino acid changes. In male crabs, the His content in testicles and TUAAs and Leu content in muscles were higher in redder hepatopancreas crabs. Amino acid levels in muscles and gonads were significantly higher in yellow hepatopancreas crabs than in white-yellow hepatopancreas crabs [14,15]. The protein of *E. sinensis* is rich in EAAs, which not only contributes to the flavor but is beneficial for human health [4,5]. In this study, ovarian TSAAs and TUAAs were lower in female crabs with a redder hepatopancreas. Therefore, the redder the hepatopancreas, the greater the nutritional quality and flavor of male muscles, while the opposite was found in female gonads. Our findings may be partially attributed to differences between the sexes. Except for the hepatopancreases, the ovaries and muscles are the main edible tissues in females and males, respectively. Additionally, even though the same aged crabs with no differences in gonad paraffin sections were selected as the experimental animals in this study, their growth and development might be inconsistent, and there were differences in fatty acid composition. There are significant differences in amino acids in muscles and gonads of *E. sinensis* sampled in different months [35]. However, further studies must be conducted to confirm our speculation.

Fatty acid composition and content are important indicators when evaluating the quality of edible aquatic species. In females, the content of MUFAs was lower in redder hepatopancreases. However, in males, the content of MUFAs, SFAs, and PUFAs was higher in redder hepatopancreases. Similarly, with the changes in hepatopancreas color (golden yellow to yellow-white to white), the proportion of SFAs and MUFAs in the hepatopancreas gradually decreased [14]. PUFAs play an important role in meat aroma. n-6 PUFAs are essential nutrients that prevent cardiovascular disease by decreasing serum cholesterol levels [36]. In this study, n-6 PUFAs of the hepatopancreases and testicles increased significantly in male crabs with a redder hepatopancreas. However, the content of EPA and DHA in the hepatopancreases and ovaries decreased in female crabs. DHA and EPA are important to human health because they inhibit the proliferation of tumor cells [37]. The differences in fatty acid content and composition might be the potential cause for the changes in carotenoids because carotenoids are liposoluble pigments. The inhibition of fatty acid metabolism is associated with a reduction in carotenoid uptake and metabolism [6]. In crabs, carotenoid extracts have high levels of unsaturated fatty acids including C16:1, C18:1, C18:3, and C20:1 [38]. Consistent with our results, these particular unsaturated fatty acids were also notably distinct in the hepatopancreas *a** of three edible parts of *E. sinensis*. Changes in fatty acids may be attributed to carotenoids. Crabs fed carotenoid-supplemented diets had a higher content of DHA, C22:4n-6, EPA, and n-3 unsaturated fatty acids in the whole body [39]. In shrimp, dietary carotenoids led to an accumulation of DHA, EPA, and n-3 PUFAs in the muscles and ovaries [40,41]. In contrast, C16:0 in muscles was decreased by dietary carotenoids [42]. In fish, the percentages of SFA and C18:3n3 decreased with increasing dietary carotenoid levels, and the percentages of C18:2n6 and n-6 PUFAs were increased by dietary carotenoids [43]. However, a study on fish (sea bream) found no impact of astaxanthin on fatty acid composition [44]. These findings indicate that the relationship between carotenoids and fatty acid composition is not only dependent on the species but on other factors.

The redder the hepatopancreas, the greater the nutritional quality and flavor of males, but not of females. Differences in the content and composition of amino acids and fatty acids in crabs with different hepatopancreas *a** values were not only dependent on the carotenoids but also influenced by other factors. Our results might be attributed to different requirements (energy and nutrients) and physiological metabolism between male and female crabs. Previous studies have shown that the fatty acid and amino acid content varied between crab tissues according to sex [45]. In our study, the fatty acid content was significantly higher in the ovaries than in the testicles. Additionally, lipid metabolism in the hepatopancreas of male crabs is more active than in female crabs during reproduction, while substance transport activity of the hepatopancreas was higher in females [45]. During maturation, female crabs accumulate nutrients in their ovaries for reproduction and successful offspring development; however, the muscles are the predominant tissue in male crabs [46]. Even though the same aged crabs had no differences in ovarian paraffin section, their growth and development might be inconsistent, with different ovarian requirements for fatty acids. Therefore, more fatty acids in the hepatopancreas might be transported and used for ovarian development in females with a redder hepatopancreas. Consistent with this hypothesis, we obtained significant increases in C18:3n3 and C20:3n3 in the ovaries of crabs with a redder hepatopancreas. Future studies should clarify the mechanism for these differences through transcriptomics, metabolomics, and genome-wide association analyses. There were positive correlations between hepatopancreas *a** values and certain nutritional components with gender-specific differences. Our findings could vary based on the species due to the distinct biochemical compositions present in various crab species. Female *Eriocheir japonica* had higher levels of PUFAs than *E. sinensis*, while *E. sinensis* had higher content of amino acid and EAA [46]. The n-3/n-6 PUFA ratio was lower in *E. sinensis* than in *Callinectes sapidus* [5,47]. Additionally, there were significant differences in mineral content between *E. sinensis* and the other crabs (*Carcinus maenus* and *Portunus pelagicus*) [5]. Further studies are needed to confirm the present findings in other crab species or even other crustaceans.

The carapaces, hepatopancreases, and ovaries of *E. sinensis* have very low color heritability, indicating no avenues for the potential genetic selective improvement of the tissue color in *E. sinensis* [23]. Local farmers should consider harvesting male crabs with a redder hepatopancreas by adjusting their diets and environmental conditions [9,48,49]. In crabs, the desired color can be achieved by supplementing the feed with astaxanthin, the carotenoid responsible for the natural color [6]. However, the cost of synthetic astaxanthin is high, which adds significantly to the costs of feed and production. Supplementing with β-carotene or products rich in β-carotene can provide a cost-effective alternative to achieve the desired color because crustaceans can convert different dietary carotenoids (including canthaxanthin, lutein, or zeaxanthin) into astaxanthin [25,50].

## 5. Conclusions

Hepatopancreas *a** was significantly positively correlated with cuirass and abdomen *a**, indicating that the hepatopancreas chroma can be assessed by examining the cuirass and abdomen. The content of total carotenoids, lutein, zeaxanthin, canthaxanthin, and β-carotene in the hepatopancreases increased significantly with increasing hepatopancreas *a** values; however, the opposite results were obtained in the ovaries. In female crabs, the content of total sugar in the hepatopancreases and ovaries increased significantly in redder hepatopancreas crabs. Interestingly, the ovarian essential amino acids (Thr and Val) and sweet amino acids (Thr, Ser, and Pro) significantly decreased in female crabs with a redder hepatopancreas. However, the His content in the testicles and the content of Leu and TUAAs in muscles increased significantly in males with a redder hepatopancreas. Consistently, the hepatopancreas DHA and EPA content significantly decreased with increasing hepatopancreas *a** values in females. In male crabs, UFA and n-6 PUFA content in the hepatopancreases and testicles increased significantly. Male crabs with a redder hepatopancreas had greater nutritional quality and flavor. Local farmers should consider harvesting male crabs with a redder hepatopancreas by adjusting their diets.

## Figures and Tables

**Figure 1 foods-13-00993-f001:**
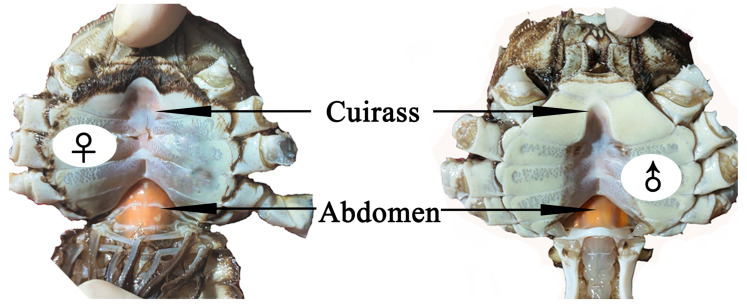
Chroma of cuirass and abdomen in *E. sinensis*.

**Figure 2 foods-13-00993-f002:**
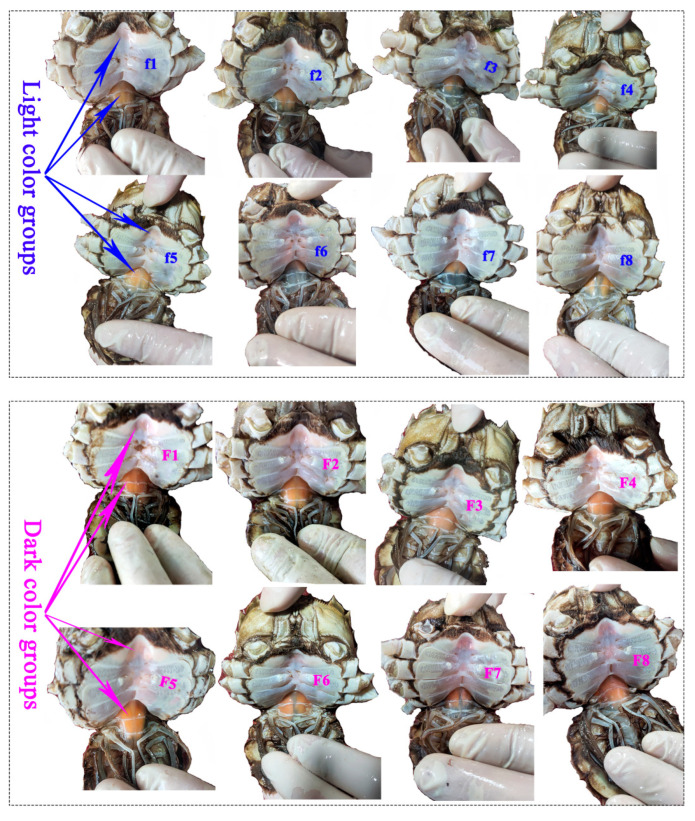
Abdomen morphology of female *E. sinensis*; f1–f8: light color groups; F1–F8: dark color groups.

**Figure 3 foods-13-00993-f003:**
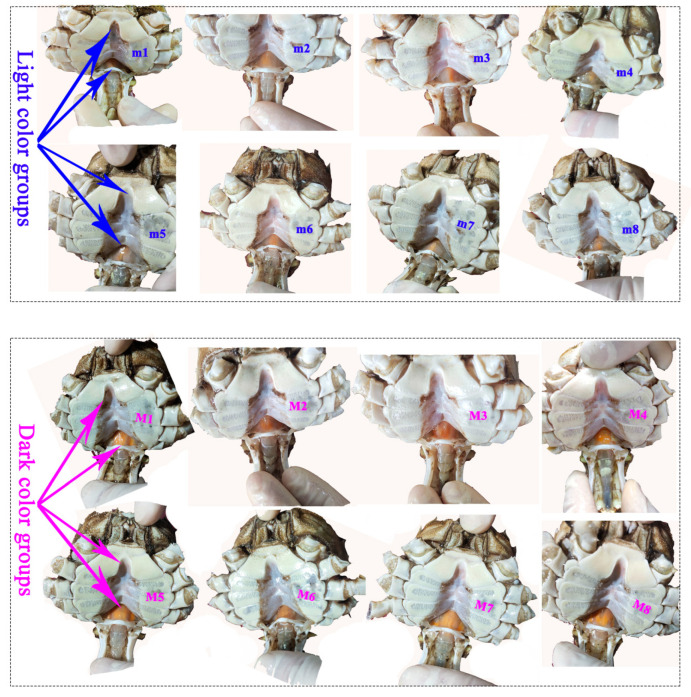
Abdomen morphology of male *E. sinensis*. m1–m8: light color groups; M1–M8: dark color group.

**Figure 4 foods-13-00993-f004:**
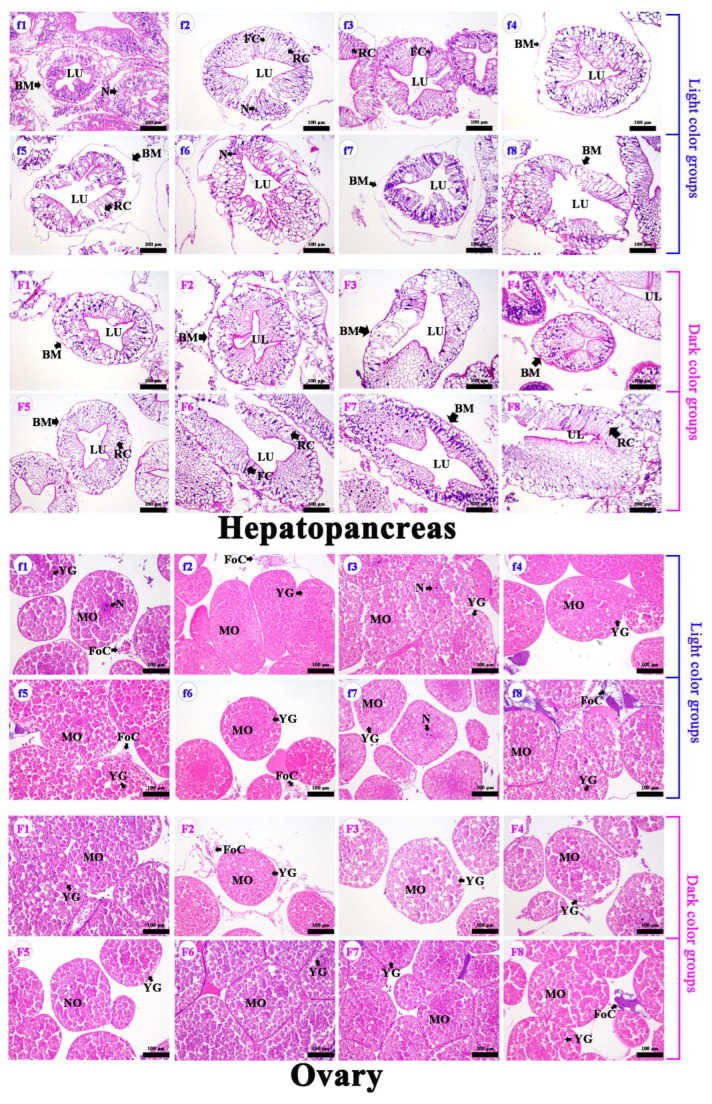
Ovarian and hepatopancreas paraffin section comparison of female *E. sinensis*; f1–f8: light color groups; F1–F8: dark color groups; MO: mature oocytes; FoC: follicle cells; YG: yolk granule; LU: lumen; BM: basement membrane; RC: resorptive cell; FC: fibrillar cell; N: nucleus.

**Figure 5 foods-13-00993-f005:**
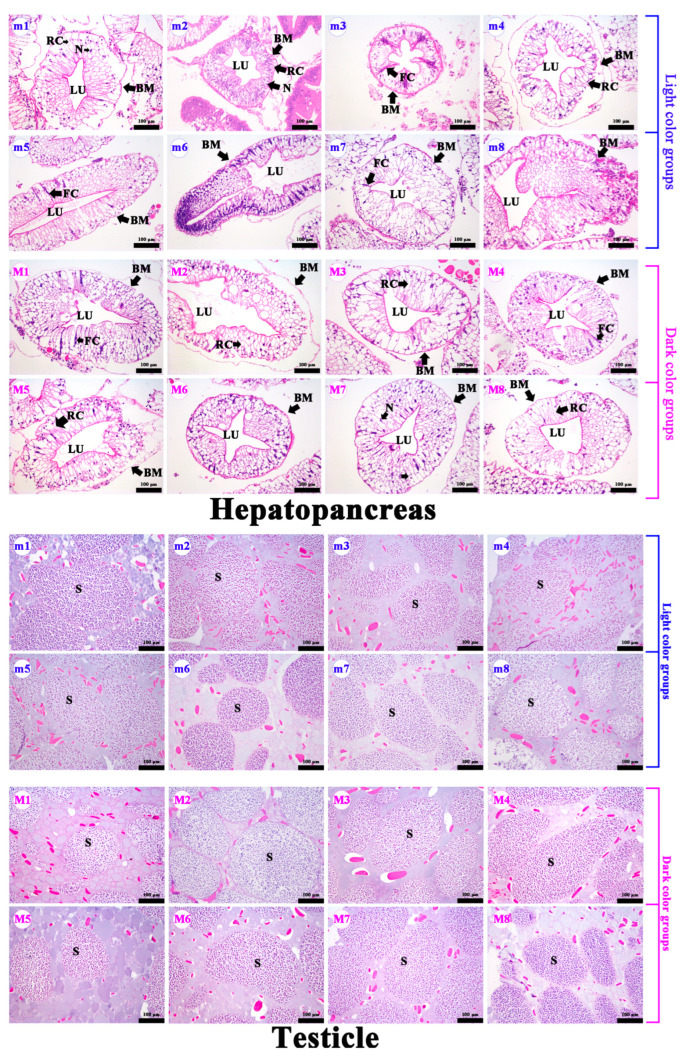
Testicle and hepatopancreas paraffin section comparison of male *E. sinensis*. m1–m8: light color groups; M1–M8: dark color groups; S: spermatid; LU: lumen; BM: basement membrane; RC: resorptive cell; FC: fibrillar cell; N: nucleus.

**Figure 6 foods-13-00993-f006:**
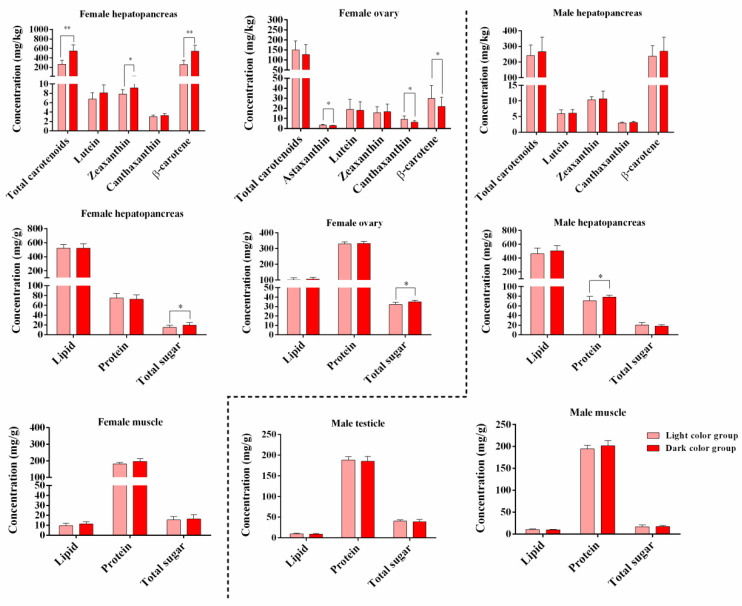
Relationships between hepatopancreas redness and carotenoids, lipids, proteins, and total sugars of *E. sinensis*. Data are presented as mean standard deviation. The values of *p* < 0.05 and 0.01 are represented as “*” and “**” above the column, indicating significant difference between the light color group and dark color group.

**Table 1 foods-13-00993-t001:** Relationships between hepatopancreas redness, body chroma, and characteristics.

Index		Females			Males		
		Light Color Group	Dark Color Group	*p*	Light Color Group	Dark Color Group	*p*
Cuirass	*L**	50.06 ± 1.63	47.87 ± 1.19	0.350	49.37 ± 2.30	49.86 ± 2.41	0.681
	*a**	5.30 ± 0.34	7.20 ± 0.66	0.039	3.14 ± 2.04	3.87 ± 1.97	0.481
	*b**	5.60 ± 0.73	4.62 ± 0.98	0.485	5.70 ± 2.78	6.06 ± 2.51	0.791
Abdomen	*L**	29.78 ± 0.88	27.88 ± 0.84	0.185	35.60 ± 2.57	32.03 ± 5.64	0.359
	*a**	5.89 ± 0.78	9.81 ± 0.85	0.009	3.11 ± 2.05	6.09 ± 3.25	0.037
	*b**	12.41 ± 1.24	13.07 ± 1.30	0.747	6.70 ± 3.08	10.95 ± 6.22	0.090
Hepatopancreas	*L**	56.81 ± 1.00	55.84 ± 0.74	0.496	57.52 ± 4.10	59.15 ± 3.07	0.639
	*a**	21.36 ± 0.94	29.55 ± 0.40	0.000	21.83 ± 1.03	26.5 ± 1.18	0.001
	*b**	55.72 ± 0.83	60.35 ± 1.12	0.010	54.03 ± 4.75	58.49 ± 3.37	0.222
	Weight (g)	8.08 ± 0.76	8.88 ± 0.87	0.547	12.36 ± 0.93	15.68 ± 2.71	0.104
	Hepatopancreas–Somatic Index (HIS/%)	7.13 ± 0.35	6.78 ± 0.41	0.574	7.38 ± 0.43	8.69 ± 0.38	0.028
Gonad	*L**	22.95 ± 0.51	23.68 ± 0.61	0.421	69.89 ± 11.56	70.01 ± 7.58	0.831
	*a**	3.96 ± 0.17	4.79 ± 0.29	0.041	−2.00 ± 0.41	−1.67 ± 0.90	0.682
	*b**	−0.69 ± 0.43	0.60 ± 0.88	0.266	−1.00 ± 0.61	−0.93 ± 0.24	0.203
	Weight (g)	12.23 ± 0.87	14.09 ± 1.37	0.323	5.56 ± 0.69	5.98 ± 0.92	0.908
	Gonad–Somatic Index (GIS/%)	10.84 ± 0.28	10.82 ± 0.38	0.971	3.33 ± 0.17	3.29 ± 0.12	0.884
Body characteristics	Weight (g)	112.73 ± 7.71	130.82 ± 12.29	0.284	173.14 ± 19.63	182.23 ± 18.22	0.928
	Height (mm)	32.37 ± 0.87	33.88 ± 1.20	0.379	35.39 ± 0.41	36.21 ± 0.49	0.509
	Carapace width (mm)	62.89 ± 1.39	65.39 ± 2.07	0.385	68.98 ± 0.81	70.41 ± 0.83	0.610
	Carapace length (mm)	57.04 ± 1.36	59.32 ± 1.81	0.385	62.44 ± 0.72	62.63 ± 0.98	0.686
	Frontal width (mm)	13.97 ± 0.40	14.34 ± 0.24	0.488	15.03 ± 0.20	15.21 ± 0.33	0.483
	First orbital margin width (mm)	36.21 ± 0.74	37.14 ± 0.86	0.479	39.21 ± 0.39	39.59 ± 0.65	0.708
	Postcarapace length (mm)	30.02 ± 0.74	31.58 ± 1.10	0.309	33.67 ± 0.47	34.12 ± 0.61	0.352
	Meropodite length of the third pereopod (mm)	39.77 ± 1.00	41.84 ± 1.47	0.315	47.75 ± 0.55	49.91 ± 0.73	0.089
	Propodite length of the third pereopod (mm)	27.87 ± 0.83	30.25 ± 1.15	0.156	33.55 ± 0.21	34.44 ± 0.50	0.197
	Dactylopodite length of the fourth pereopod (mm)	21.20 ± 0.51	21.93 ± 0.85	0.520	24.39 ± 0.50	24.42 ± 0.71	0.361

Note: Data expressed as mean ± standard deviation.

**Table 2 foods-13-00993-t002:** Relationships between hepatopancreas redness and amino acid content (mg/g of wet tissue weight).

	Females									Males								
	Hepatopancreas			Ovary			Muscle			Hepatopancreas			Testicle			Muscle		
	Light Color Groups	Dark Color Groups	*p*	Light Color Groups	Dark Color Groups	*p*	Light Color Groups	Dark Color Groups	*p*	Light Color Groups	Dark Color Groups	*p*	Light Color Groups	Dark Color Groups	*p*	Light Color Groups	Dark Color Groups	*p*
Aspartic acid (Asp) +	0.56 ± 0.39	0.51 ± 0.33	0.780	0.91 ± 0.09	0.62 ± 0.23	0.008	0.09 ± 0.07	0.08 ± 0.03	0.832	0.91 ± 0.35	0.77 ± 0.47	0.514	1.69 ± 1.06	2.30 ± 1.55	0.372	0.08 ± 0.02	0.10 ± 0.02	0.087
Threonine (Thr) #,*	1.49 ± 0.25	1.45 ± 0.21	0.750	1.06 ± 0.13	0.81 ± 0.21	0.012	0.40 ± 0.25	0.32 ± 0.09	0.482	1.71 ± 0.32	1.59 ± 0.32	0.448	0.11 ± 0.04	0.17 ± 0.08	0.069	0.40 ± 0.16	0.40 ± 0.11	0.967
Serine (Ser) *	1.01 ± 0.28	0.99 ± 0.15	0.888	1.26 ± 0.14	1.05 ± 0.21	0.032	0.46 ± 0.26	0.49 ± 0.16	0.866	1.23 ± 0.28	1.08 ± 0.31	0.350	0.19 ± 0.08	0.30 ± 0.14	0.102	0.45 ± 09.26	0.52 ± 0.07	0.537
Glutamic acid (Glu) +	2.33 ± 0.47	2.26 ± 0.33	0.762	2.19 ± 0.29	1.71 ± 0.34	0.010	1.03 ± 0.73	0.56 ± 0.12	0.178	2.69 ± 0.52	2.40 ± 0.56	0.302	1.44 ± 0.64	1.85 ± 1.06	0.373	0.54 ± 0.30	0.82 ± 0.19	0.093
Glycine (Gly) *	1.09 ± 0.28	1.08 ± 0.18	0.926	0.63 ± 0.06	0.52 ± 0.08	0.006	3.28 ± 1.02	3.88 ± 1.35	0.404	1.19 ± 0.27	1.17 ± 0.27	0.869	0.45 ± 0.15	0.45 ± 0.13	1.000	4.28 ± 0.84	5.42 ± 1.30	0.103
Cystine (Cys)	0.25 ± 0.08	0.27 ± 0.05	0.545	0.04 ± 0.02	0.03 ± 0.02	0.297	0.04 ± 0.02	0.03 ± 0.01	0.364	0.36 ± 0.07	0.34 ± 0.07	0.659	0.05 ± 0.03	0.09 ± 0.08	0.239	0.04 ± 0.01	0.04 ± 0.02	0.380
Valine (Val) #	1.61 ± 0.32	1.54 ± 0.25	0.612	1.33 ± 0.15	0.98 ± 0.34	0.020	0.30 ± 0.20	0.23 ± 0.08	0.442	1.91 ± 0.38	1.78 ± 0.34	0.459	0.13 ± 0.06	0.18 ± 0.09	0.148	0.26 ± 0.11	0.30 ± 0.08	0.458
Methionine (Met) #	0.67 ± 0.19	0.59 ± 0.13	0.331	1.13 ± 0.29	0.86 ± 0.26	0.069	0.24 ± 0.13	0.15 ± 0.05	0.140	0.77 ± 0.20	0.69 ± 0.17	0.404	<0.0075	<0.0075	-	0.13 ± 0.06	0.19 ± 0.06	0.106
Isoleucine (Ile) #	0.85 ± 0.38	0.82 ± 0.35	0.909	0.65 ± 0.25	0.47 ± 0.23	0.169	0.09 ± 0.06	0.07 ± 0.03	0.703	1.24 ± 0.38	1.00 ± 0.31	0.198	0.13 ± 0.05	0.17 ± 0.10	0.392	0.10 ± 0.04	0.12 ± 0.05	0.467
Leucine (Leu) #	1.64 ± 0.84	1.47 ± 0.65	0.661	2.56 ± 1.08	2.04 ± 0.95	0.320	0.22 ± 0.14	0.20 ± 0.09	0.762	2.33 ± 0.79	1.91 ± 0.65	0.269	0.15 ± 0.06	0.21 ± 0.14	0.242	0.18 ± 0.08	0.32 ± 0.13	0.040
Tyrosine (Tyr)	0.67 ± 0.54	0.45 ± 0.24	0.304	1.23 ± 0.80	1.12 ± 0.71	0.781	0.22 ± 0.08	0.11 ± 0.00	0.304	1.07 ± 0.52	0.74 ± 0.36	0.158	<0.0095	<0.0095	-	<0.0095	0.19 ± 0.09	0.921
Phenylalanine (Phe) #,*	0.23 ± 0.31	0.17 ± 0.15	0.642	0.34 ± 0.35	0.54 ± 0.61	0.439	0.13 ± 0.05	0.09 ± 0.00	0.722	0.22 ± 0.16	0.23 ± 0.15	0.929	<0.0083	<0.0083	-	<0.0083	0.19 ± 0.07	-
Lysine (Lys) #	2.29 ± 0.44	2.11 ± 0.37	0.400	1.24 ± 0.32	1.00 ± 0.26	0.119	0.60 ± 0.34	0.33 ± 0.11	0.119	2.63 ± 0.55	2.33 ± 0.47	0.261	0.23 ± 0.10	0.31 ± 0.16	0.239	0.47 ± 0.22	0.37 ± 0.07	0.333
Histidine (His) #	0.70 ± 0.13	0.67 ± 0.11	0.566	0.60 ± 0.09	0.51 ± 0.11	0.088	0.28 ± 0.14	0.12 ± 0.05	0.111	0.84 ± 0.17	0.77 ± 0.16	0.402	0.09 ± 0.03	0.15 ± 0.06	0.044	0.19 ± 0.07	0.19 ± 0.04	1.000
Arginine (Arg)	2.41 ± 0.72	2.39 ± 0.69	0.945	3.04 ± 0.57	2.86 ± 0.42	0.495	3.42 ± 1.48	4.98 ± 1.49	0.099	3.10 ± 0.89	3.01 ± 0.79	0.839	0.36 ± 0.20	0.53 ± 0.10	0.053	3.83 ± 1.47	5.37 ± 0.58	0.039
Proline (Pro)	2.13 ± 0.46	2.05 ± 0.37	0.722	2.18 ± 0.32	1.68 ± 0.31	0.006	3.08 ± 1.44	2.90 ± 1.12	0.821	2.70 ± 0.61	2.78 ± 0.68	0.819	0.54 ± 0.31	0.62 ± 0.19	0.574	1.83 ± 0.65	2.33 ± 0.25	0.111
TAAs	22.54 ± 4.91	21.60 ± 3.74	0.674	22.20 ± 3.79	18.44 ± 4.41	0.089	17.48 ± 4.60	18.77 ± 5.31	0.664	27.70 ± 6.03	25.51 ± 5.48	0.460	7.19 ± 2.86	9.56 ± 4.59	0.234	18.25 ± 3.27	21.50 ± 3.00	0.103
EAAs	8.74 ± 2.35	8.13 ± 1.88	0.572	8.30 ± 2.29	6.69 ± 2.35	0.188	1.84 ± 1.04	1.31 ± 0.47	0.287	10.81 ± 2.67	9.52 ± 2.23	0.313	0.74 ± 0.28	1.06 ± 0.54	0.171	1.54 ± 0.62	1.80 ± 0.56	0.457
TUAAs	2.89 ± 0.80	2.78 ± 0.49	0.738	3.10 ± 0.36	2.34 ± 0.56	0.005	1.11 ± 0.78	0.63 ± 0.15	0.198	3.60 ± 0.70	3.17 ± 0.92	0.313	3.13 ± 1.68	4.15 ± 2.59	0.369	0.62 ± 0.31	0.91 ± 0.20	0.048
TSAAs	8.47 ± 1.29	8.38 ± 1.06	0.881	6.99 ± 0.76	5.65 ± 0.86	0.005	11.08 ± 3.42	11.89 ± 3.41	0.694	9.60 ± 1.85	9.54 ± 2.02	0.947	2.94 ± 1.06	3.77 ± 1.45	0.216	11.84 ± 2.55	13.45 ± 1.81	0.236

Note: “#” represents essential amino acids; “+” represents umami amino acids; “*” represents sweet amino acids. Data are expressed as mean ± standard deviation. “-” indicates that it was not detected. TTAs: total amino acids; EAAs: essential amino acids; TUAAs: total umami amino acids; TSAAs: total sweet amino acids.

**Table 3 foods-13-00993-t003:** Relationships between hepatopancreas redness and fatty acid composition.

	Females									Males								
	Hepatopancreas			Ovary			Muscle			Hepatopancreas			Testicle			Muscle		
	Light Color Group	Dark Color Group	*p*	Light Color Group	Dark Color Group	*p*	Light Color Group	Dark Color Group	*p*	Light Color Group	Dark Color Group	*p*	Light Color Group	Dark Color Group	*p*	Light Color Group	Dark Color Group	*p*
Saturated fatty acids (SFAs, g/kg tissue weight)																		
C6:0	<0.0033	<0.0033	-	<0.0033	<0.0033	-	<0.0033	<0.0033	-	0.04 ± 0.06	0.12 ± 0.00	0.449	<0.0033	<0.0033	-	<0.0033	<0.0033	-
C10:0	0.13 ± 0.05	0.14 ± 0.06	1.000	<0.0066	<0.0066	-	<0.0066	<0.0066	-	0.12 ± 0.06	0.15 ± 0.08	0.451	<0.0066	<0.0066	-	<0.0066	<0.0066	-
C12:0	11.04 ± 4.73	10.43 ± 5.18	0.807	0.43 ± 0.15	0.43 ± 0.15	0.740	<0.0066	<0.0066	-	10.56 ± 2.54	14.07 ± 5.85	0.143	<0.0066	0.10 ± 0.00	-	<0.0033	<0.0033	-
C13:0	0.14 ± 0.02	0.14 ± 0.03	0.865	<0.0033	<0.0033	-	<0.0033	<0.0033	-	0.12 ± 0.02	0.14 ± 0.03	0.138	<0.0033	<0.0033	-	<0.0033	<0.0033	-
C14:0	6.42 ± 1.25	5.60 ± 1.44	0.245	0.92 ± 0.18	0.84 ± 0.17	0.378	<0.0033	<0.0033	-	5.39 ± 0.56	6.73 ± 1.84	0.083	<0.0033	0.05 ± 0.00	-	<0.0033	<0.0033	-
C15:0	0.92 ± 0.23	0.89 ± 0.18	0.795	0.16 ± 0.03	0.19 ± 0.04	0.103	<0.0033	<0.0033	-	0.69 ± 0.12	0.88 ± 0.13	0.010	<0.0033	<0.0033	-	<0.0033	<0.0033	-
C16:0	62.21 ± 9.77	56.19 ± 7.11	0.181	11.52 ± 1.61	12.01 ± 1.58	0.547	0.48 ± 0.08	0.53 ± 0.08	0.342	46.90 ± 6.86	57.27 ± 9.07	0.022	0.31 ± 0.12	0.34 ± 0.10	0.541	0.43 ± 0.07	0.47 ± 0.08	0.340
C17:0	0.68 ± 0.16	0.60 ± 0.11	0.281	0.22 ± 0.04	0.25 ± 0.06	0.293	<0.0066	<0.0066	-	0.58 ± 0.11	0.64 ± 0.11	0.311	<0.0066	<0.0066	-	<0.0066	<0.0066	-
C18:0	9.08 ± 1.90	7.88 ± 1.81	0.218	3.95 ± 0.64	3.96 ± 0.66	0.979	0.35 ± 0.08	0.34 ± 0.04	0.782	7.28 ± 0.94	8.13 ± 1.57	0.206	0.31 ± 0.12	0.26 ± 0.04	0.295	0.37 ± 0.04	0.37 ± 0.05	0.974
C20:0	0.62 ± 0.13	0.55 ± 0.14	0.331	0.21 ± 0.03	0.21 ± 0.04	0.948	<0.0066	<0.0066	-	0.52 ± 0.09	0.60 ± 0.10	0.123	<0.0066	0.07 ± 0.00	-	<0.0033	<0.0033	-
C21:0	0.24 ± 0.02	0.28 ± 0.03	0.204	<0.0033	<0.0033	-	<0.0033	<0.0033	-	0.09 ± 0.12	0.13 ± 0.17	0.611	<0.0033	<0.0033	-	<0.0033	<0.0033	-
C22:0	0.52 ± 0.13	0.44 ± 0.15	0.288	0.10 ± 0.02	0.11 ± 0.03	0.414	0.07 ± 0.00	<0.0033	-	0.48 ± 0.10	0.55 ± 0.14	0.294	0.02 ± 0.04	0.02 ± 0.04	0.966	<0.0033	<0.0033	-
C23:0	<0.0033	0.24 ± 0.05	-	<0.0033	<0.0033	-	<0.0033	<0.0033	-	0.08 ± 0.11	0.11 ± 0.15	0.694	<0.0033	<0.0033	-	<0.0033	<0.0033	-
C24:0	<0.0066	0.14 ± 0.09	-	<0.0066	<0.0066	-	<0.0066	<0.0066	-	0.04 ± 0.07	0.08 ± 0.11	0.408	<0.0066	<0.0066	-	<0.0033	<0.0033	-
SFAs	91.83 ± 11.90	83.00 ± 12.75	0.174	17.53 ± 2.46	18.00 ± 2.43	0.708	0.85 ± 0.17	0.87 ± 0.11	0.870	72.87 ± 8.50	89.49 ± 15.93	0.021	0.64 ± 0.24	0.65 ± 0.16	0.876	0.82 ± 0.20	0.84 ± 0.13	0.826
Monounsaturated fatty acids (MUFAs, g/kg tissue weight)																		
C14:1	1.11 ± 0.19	0.96 ± 0.20	0.141	0.07 ± 0.01	0.06 ± 0.01	0.201	<0.0033	<0.0033	-	0.70 ± 0.15	0.94 ± 0.23	0.027	<0.0033	<0.0033	-	<0.0033	<0.0033	-
C16:1	32.25 ± 5.10	30.13 ± 6.51	0.480	9.05 ± 1.67	9.23 ± 2.23	0.858	0.09 ± 0.05	0.16 ± 0.07	0.050	20.98 ± 6.09	27.39 ± 7.10	0.073	0.05 ± 0.04	0.09 ± 0.04	0.050	0.07 ± 0.03	0.10 ± 0.06	0.250
C18:1n9c	115.51 ± 17.37	97.70 ± 11.93	0.031	24.81 ± 3.17	23.81 ± 3.34	0.549	0.78 ± 0.14	0.83 ± 0.12	0.530	90.59 ± 12.72	108.33 ± 16.56	0.031	0.46 ± 0.13	0.61 ± 0.21	0.096	0.65 ± 0.12	0.70 ± 0.10	0.421
C20:1	2.64 ± 0.81	2.12 ± 0.45	0.135	0.40 ± 0.07	0.39 ± 0.10	0.954	0.04 ± 0.00	<0.0033	-	1.60 ± 0.31	1.79 ± 0.33	0.254	0.01 ± 0.02	0.02 ± 0.02	0.297	<0.0033	<0.0033	-
C22:1n9	0.18 ± 0.06	0.14 ± 0.07	0.237	0.14 ± 0.02	0.16 ± 0.05	0.361	0.04 ± 0.00	0.03 ± 0.00	0.212	0.17 ± 0.04	0.17 ± 0.07	0.986	0.03 ± 0.02	0.02 ± 0.03	0.596	<0.0033	<0.0033	-
C24:1	<0.0033	<0.0033	-	<0.0033	<0.0033	-	<0.0033	<0.0033	-	<0.0033	<0.0033	-	0.35 ± 0.18	0.37 ± 0.23	0.853	<0.0033	<0.0033	-
MUFAs	151.69 ± 22.23	131.04 ± 11.29	0.034	34.47 ± 4.48	33.65 ± 4.54	0.724	0.89 ± 0.19	1.00 ± 0.17	0.320	114.04 ± 17.43	138.63 ± 20.40	0.021	0.89 ± 0.28	1.11 ± 0.37	0.201	0.71 ± 0.14	0.80 ± 0.15	0.292
Polyunsaturated fatty acids (PUFAs, g/kg tissue weight)																		
C18:2n6c	60.41 ± 8.36	58.07 ± 17.28	0.735	16.74 ± 2.35	17.43 ± 2.99	0.617	0.56 ± 0.12	0.70 ± 0.12	0.078	57.79 ± 9.58	73.83 ± 15.85	0.028	0.23 ± 0.09	0.37 ± 0.15	0.037	0.57 ± 0.15	0.66 ± 0.11	0.268
C18:3n6	0.08 ± 0.02	0.10 ± 0.02	0.105	<0.0066	<0.0066	-	<0.0066	<0.0066	-	0.03 ± 0.04	0.06 ± 0.05	0.292	<0.0033	<0.0033	-	<0.0033	<0.0033	-
C18:3n3	6.70 ± 1.74	7.32 ± 1.52	0.464	2.38 ± 0.40	3.40 ± 0.48	0.000	0.06 ± 0.02	0.11 ± 0.03	0.011	4.91 ± 1.32	6.61 ± 1.70	0.042	0.004 ± 0.01	0.03 ± 0.02	0.007	0.06 ± 0.01	0.09 ± 0.03	0.041
C20:2	6.01 ± 1.41	4.63 ± 1.13	0.049	1.23 ± 0.19	1.33 ± 0.28	0.388	0.11 ± 0.03	0.09 ± 0.01	0.205	4.30 ± 1.03	4.44 ± 0.97	0.784	0.10 ± 0.02	0.11 ± 0.02	0.561	0.10 ± 0.01	0.09 ± 0.02	0.702
C20:3n6	0.25 ± 0.11	0.19 ± 0.12	0.395	0.09 ± 0.03	0.10 ± 0.05	0.598	<0.0033	<0.0033	-	0.17 ± 0.05	0.23 ± 0.09	0.089	<0.0033	<0.0033	-	<0.0033	<0.0033	-
C20:3n3	1.62 ± 0.44	1.53 ± 0.32	0.661	0.37 ± 0.07	0.49 ± 0.12	0.032	<0.0033	0.04 ± 0.00	-	0.99 ± 0.13	1.18 ± 0.28	0.106	0.05 ± 0.03	0.06 ± 0.04	0.634	<0.0033	<0.0033	-
C20:4n6	2.90 ± 0.56	2.22 ± 0.64	0.041	2.48 ± 0.48	2.13 ± 0.65	0.247	0.27 ± 0.08	0.23 ± 0.05	0.266	2.78 ± 0.70	2.92 ± 0.85	0.714	0.35 ± 0.12	0.43 ± 0.06	0.111	0.23 ± 0.02	0.24 ± 0.02	0.371
C20:5n3 (EPA)	2.27 ± 0.60	1.65 ± 0.47	0.038	3.15 ± 0.58	2.47 ± 0.56	0.034	0.47 ± 0.09	0.41 ± 0.04	0.199	2.49 ± 0.53	2.59 ± 0.84	0.789	0.18 ± 0.07	0.21 ± 0.03	0.241	0.43 ± 0.21	0.46 ± 0.07	0.764
C22:2	<0.0033	<0.0033	-	<0.0033	<0.0033	-	<0.0033	<0.0033	-	<0.0033	0.03 ± 0.06	-	<0.0033	<0.0033	-	<0.0033	<0.0033	-
C22:6n3 (DHA)	1.28 ± 0.31	0.85 ± 0.38	0.029	2.00 ± 0.54	1.56 ± 0.79	0.214	0.25 ± 0.05	0.25 ± 0.06	0.869	1.29 ± 0.30	1.21 ± 0.40	0.689	0.08 ± 0.04	0.10 ± 0.02	0.095	0.31 ± 0.05	0.26 ± 0.06	0.197
PUFA	81.49 ± 10.67	76.49 ± 18.90	0.098	28.44 ± 3.30	28.93 ± 4.47	0.935	1.72 ± 0.25	1.80 ± 0.18	0.526	74.75 ± 10.73	93.12 ± 18.08	0.027	0.99 ± 0.27	1.31 ± 0.21	-	1.69 ± 0.29	1.80 ± 0.22	0.484
n-3 PUFAs	11.87 ± 2.66	11.35 ± 2.28	0.684	7.90 ± 1.06	7.92 ± 1.43	0.966	0.78 ± 0.12	0.79 ± 0.08	0.856	9.68 ± 1.26	11.61 ± 2.37	0.061	0.31 ± 0.11	0.40 ± 0.08	0.073	0.80 ± 0.24	0.81 ± 0.12	0.914
n-6 PUFAs	63.61 ± 8.50	60.51 ± 17.27	0.656	19.32 ± 2.63	19.67 ± 3.28	0.815	0.83 ± 0.14	0.92 ± 0.13	0.264	60.77 ± 9.56	77.36 ± 16.08	0.025	0.58 ± 0.14	0.80 ± 0.14	0.009	0.79 ± 0.15	0.10 ± 0.10	0.208
n-3 PUFAs/ n-6 PUFAs	0.19 ± 0.03	0.20 ± 0.06	0.680	0.41 ± 0.06	0.41 ± 0.07	0.895	0.94 ± 0.12	0.86 ± 0.11	0.289	0.16 ± 0.01	0.15 ± 0.02	0.448	0.52 ± 0.10	0.51 ± 0.09	0.785	1.04 ± 0.34	0.91 ± 0.11	0.396
UFAs	233.18 ± 29.60	207.53 ± 28.26	0.098	62.91 ± 7.45	62.58 ± 8.28	0.807	2.61 ± 0.40	2.80 ± 0.33	0.382	188.79 ± 26.62	231.75 ± 37.41	0.019	1.88 ± 0.49	2.42 ± 0.57	0.043	2.40 ± 0.36	2.61 ± 0.34	0.342
Total fatty acids	325.00 ± 41.88	290.54 ± 47.46	0.146	80.45 ± 9.76	80.58 ± 10.56	0.981	3.45 ± 0.55	3.67 ± 0.46	0.478	261.65 ± 34.53	321.23 ± 52.12	0.017	2.55 ± 0.47	3.08 ± 0.70	0.095	3.28 ± 0.43	3.45 ± 0.45	0.525

Note: Data are expressed as mean ± standard deviation. “-” indicates that it was not detected.

## Data Availability

The original contributions presented in the study are included in the article/Appendix A, further inquiries can be directed to the corresponding author.

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
