# Peer review of "Comparison of Body Characteristics, Carotenoid Composition, and Nutritional Quality of Chinese Mitten Crab (Eriocheir sinensis) with Different Hepatopancreas Redness"

_foods, 2024, doi:10.3390/foods13070993_

Round 1

Reviewer 1 Report (Previous Reviewer 1)

Comments and Suggestions for Authors

Points Still to be Addressed

Abstract:

Lack of a clear and concise statement of the research objective. Repetition of information in different sections.

Results Section:

Lack of clarity on significance levels in statistical results. Need for discussion on the biological significance of reported statistical differences. Absence of a clear interpretation of results aligning with study objectives. Lack of a concluding section that synthesizes findings and discusses broader implications.

Discussion Section:

Failure to acknowledge existing gaps in the literature or potential limitations. Insufficient exploration of potential limitations in generalizing findings to other species. Lack of discussion on potential confounding factors or alternative explanations for observed relationships. Need for transparent discussion on limitations or assumptions in statistical analyses. Limited exploration of practical applications or implications for the seafood industry, consumers, or resource management.

Comments on the Quality of English Language

Moderate editing of English language required

Author Response

Dear Sir or Madam:

Thank you for your comments concerning our manuscript entitled “Comparison of body characteristics, carotenoids composition and nutritional quality in Chinese mitten crab (Eriocheir sinensis) with different hepatopancreases redness” (foods-2889156). Those comments are all valuable and very helpful for revising and improving our paper, as well as the important guiding significance to our researches. We have studied comments carefully and have made correction which we hope meet with approval. The main corrections and responds are as following 

Abstract:

  • Lack of a clear and concise statement of the research objective.

Thank you. We have rewritten the statement of the research objective according to your suggestion as follows, “The objective of this study was to investigate the body characteristics, carotenoids composition, and nutritional quality of Eriocheir sinensis with different hepatopancreas redness (a*)”. We hope that our revisions are proper and satisfactory.

  • Repetition of information in different sections.

In our revised manuscript, the repetitive information was deleted according to your suggestion.

Results Section:

  • Lack of clarity on significance levels in statistical results.

Thank you. According to your suggestion, the significant levels of our results were clarified in the revised manuscript.

  • Need for discussion on the biological significance of reported statistical differences.

Thank you. We provided the biological significance of our significant differences in our revised manuscript.

  • Absence of a clear interpretation of results aligning with study objectives.

Thank you. We provided some reasonably interpretations for the significant differences between the different groups according to your suggestion.

  • Lack of a concluding section that synthesizes findings and discusses broader implications.

In the Result Section, we provided concluding sections of synthesizes findings and discusses according to your suggestion.

Discussion Section:

  • Failure to acknowledge existing gaps in the literature or potential limitations.

Thank you. It is well known that the hepatopancreas were the mainly edible parts of Chinese mitten crab with high nutrition. Some previous researches had revealed that the chroma of carapace, hepatopancreas and gonad have significantly relationships to regions, size, ages and culture modes (Zhang et al., 2020; Li et al., 2020). In addition, some researches were focused on the relationship of hepatopancreas yellowness to nutritious quality and health in recent years (Wang et al., 2021a,b). However, there were limited researches reporting the relationship between the hepatopancreas redness and qualities of E. sinensis. However, there is no standard color for E. sinensis, it is traditionally held that "redder" hepatopancreas are of superior quality.  Therefore, the consumer studies and market analysis data should be needed to solidify consumer behavior and preference claims in the future. Additionally, it must be clarified that there might be limitations on our results due to the crabs were sampled from one farming, and they belonged the same breed. The further researches are needed for discussing other crab species or even other crustaceans. Moreover, the contents and types of carotenoids in crabs were usually affected by diets and environment.  The pond culture enhanced the levels of protein, amino acids, and specific organic acid derivatives, and reduced the levels of peptides and polyunsaturated fatty acids (PUFAs) compared with crabs in lake (Ye et al., 2023). The contents of carotenoids, DHA, EPA, EAA, and FAA were improved significantly in male hepatopancreas and muscle under saline water and alkaline water (Wang et al., 2024). Therefore, reliability of hepatopancreas redness as an indicator of nutritional quality would be strengthen under different culture modes. We have clarified this perspective in the Discussion section of revised manuscript.

References:

Zhang, L.; Yin, M.Y.; & Wang, X.C. Meat texture, muscle histochemistry and protein composition of Eriocheir sinensis with different size traits. Food Chemistry, 2020, 338(Feb.15), 127632. https://doi.org/10.1016/j.foodchem.2020.127632

Li, Q.Q.; Zu, L.; Cheng, Y.X.; Wade, N.M.; Liu, J.G.; & Wu, X.G. Carapace color affects carotenoid composition and nutritional quality of the Chinese mitten crab, Eriochier sinensis. Lwt-Food Science And Technology, 2020, 126. https://doi.org/10.1016/j.lwt.2020.109286

Wang, Q.J.; Jiang, X.D.; Yao, Q.; Long, X.W.; Xu, Y.P.; Wu, M.; & Zhang, D.M. Comparative study on the nutrition composition of adult male Chinese mitten crab (Eriocheir sinensis) with different coloured hepatopancreases. Aquaculture Research,  2021a, 52(1), 196-207. https://doi.org/10.1111/are.14881

Wang, Q.J.; Zhang, B.Y.; Jiang, X.D.; Long, X.W.; Zhu, W.L.; Xu, Y.P.; Wu, M.; & Zhang, D.M. Comparison on nutritional quality of adult female Chinese mitten crab (Eriocheir sinensis) with different colored hepatopancreases. Journal of Food Science, 2021b, 86(5), 2075-2090. https://doi.org/10.1111/1750-3841.15664.

Ye, Y.; Wang, Y.; Liu, P.; Chen, J.; & Zhang, C. Uncovering the nutritive profiles of adult male chinese mitten crab (E. sinensis) harvested from the pond and natural water area of Qin lake based on metabolomics. Foods (Basel, Switzerland), 2023, 12(11), 2178. https://doi.org/10.3390/foods12112178

Wang, S.; Guo, K.; Luo, L.; Zhang, R.; Xu, W.; Song, Y.; & Zhao, Z. Fattening in saline and alkaline water improves the color, nutritional and taste quality of adult Chinese mitten crab Eriocheir sinensis. Foods (Basel, Switzerland), 2022, 11(17), 2573. https://doi.org/10.3390/foods11172573

  • Insufficient exploration of potential limitations in generalizing findings to other species.

In our present study, Spearman correlation analysis shown that carotenoids levels had a positive correlation with hepatopancreas a* values. Similarly, The β-carotene in the brown hepatopancreas group was significantly lower than that in the orange hepatopancreas (Zhang et al., 2022). In shrimp, there was a 2.4-fold difference in total carotenoids content of shell extracts of reds compared to whites (Wade et al., 2005). The redness (a*) values, total astaxanthin and astaxanthin ester contents of the carapace, hepatopancreas, and ovaries of purple crabs was significantly higher than those of greenblack crabs (Li et al., 2020). Therefore, the scoring of crustacean redness could be evaluated through measuring the accurate carotenoids contents in future researches. Additionally, the differences of fatty acid contents and composition may be attributed to carotenoids in our study. Similarly, crabs fed with carotenoid-supplemented diets showed higher contents of DHA, C22:4n-6 and EPA, and n-3 highly unsaturated fatty acid in whole body (Han et al., 2018). In shrimps, dietary carotenoids led to an accumulation of DHA, EPA, and n-3 PUFA in muscle or ovary (Paibulkichakul et al., 2008; Wang et al., 2018). On the contrary, a decline of palmitic acid (C16:0) proportion in muscle was decreased by dietary carotenoids (Chen et al., 2023). In fishes, percentages of SFA and C18:3n3 decreased with increasing dietary carotenoids levels, and the percentages of C18:2n6 and n-6 PUFA increased by dietary carotenoids (Long et al., 2023). However, another study on fish (sea bream) found no impact of astaxanthin on fatty acids composition (Tejera et al., 2007). Additionally, our conclusion might be species-specific owing to different biochemical composition among different crabs. Taken together, these findings indicate that the relationship between carotenoids and fatty acids composition is not only dependent on the species but also influenced by other factors. According to your suggestion, we have supplied the information for exploring limitations of our finding to other species in our revised manuscript.

Reference:

Wade, N., Goulter, K.C., Wilson, K.J., Hall, M.R., & Degnan, B.M. Esterified astaxanthin levels in lobster epithelia correlate with shell colour intensity: potential role in crustacean shell colour formation. Comparative biochemistry and physiology. Part B, Biochemistry & molecular biology, 2005, 141 3, 307-13.

Zhang, L.; Tao, N.P.; Wu, X.G.; & Wang, X.C. Metabolomics of the hepatopancreas in Chinese mitten crabs (Eriocheir sinensis). Food Res Int, 2022, 152, 110914.

Li, Q.Q.; Zu, L.; Cheng, Y.X.; Wade, N.M.; Liu, J.G.; & Wu, X.G. Carapace color affects carotenoid composition and nutritional quality of the Chinese mitten crab, Eriochier sinensis. Lwt-Food Sci Technol, 2020, 126.

Paibulkichakul, C., Piyatiratitivorakul, S., Sorgeloos, P., & Menasveta, P. (2008). Improved maturation of pond-reared, black tiger shrimp (Penaeus monodon) using fish oil and astaxanthin feed supplements. Aquaculture, 282, 83-89.

Wang, W., Ishikawa, M., Koshio, S., Yokoyama, S., Hossain, M.S., & Moss, A.S. Effects of dietary astaxanthin supplementation on juvenile kuruma shrimp, Marsupenaeus japonicus. Aquaculture, 491, 2018, 197-204.

Han, T., Li, X., Wang, J., Wang, C., Yang, M., & Zheng, P. Effects of dietary astaxanthin (AX) supplementation on pigmentation, antioxidant capacity and nutritional value of swimming crab, Portunus trituberculatus. Aquaculture, 2018, 490, 169-177.

Chen, Q., Huang, S., Dai, J., Wang, C., Chen, S., Qian, Y., Gong, Y., & Han, T. Effects of synthetic astaxanthin on the growth performance, pigmentation, antioxidant capacity, and immune response in black tiger prawn (Penaeus monodon). Aquaculture nutrition, 2023, 6632067.

Long, X., Wang, L., Li, Y., Sun, W., & Wu, X. Effects of long-term Haematococcus pluvialis astaxanthin feeding on the growth, coloration, and antioxidant capacity of commercial-sized Oncorhynchus mykiss. Aquaculture Reports, 2023.

Tejera, J.R. Cejas, C. Rodríguez, B. Bjerkeng, S. Jerez, A. Bolaños, A. Lorenzo Pigmentation, carotenoids, lipid peroxides and lipid composition of skin of red porgy (Pagrus pagrus) fed diets supplemented with different astaxanthin sources Aquaculture, 270 (1–4) (2007), pp. 218-230.

  • Lack of discussion on potential confounding factors or alternative explanations for observed relationships.

In our present study, the redder hepatopancreas, the better nutritional quality and flavor of male E. sinensis, but not females. Apparently, differences on the contents and composition of amino and fatty acids in the crabs with different hepatopancreas a* were not only dependent on the carotenoids but also influenced by other factors. Our results might be attributed to different requirements (energy and nutrient) and physiological metabolism between male and female crabs. Previous studies have proven the fatty acid and amino acid contents varied between crabs tissues according to sex (Barrento et al., 2009; Czerniejewski et al., 2023). Additionally, lipid metabolism in the hepatopancreas of male crabs is more active than in that of female crabs during the reproduction, while substance transportation activity of hepatopancreas was higher in females. During maturation, female crabs accumulate nutrients in their ovaries for reproduction and successful offspring development, however the muscles are the predominant tissue in male crabs. On the other hand, although the same aged crabs were no differences on ovarian paraffin section in this study, their growth and development might be inconsistent, and there were different requirements of fatty acids in ovary. According to your suggestion, we have supplied some related discussions to clarify our current results in our revised manuscript.

Reference:

Barrento, S.I.; Marques, A.; Teixeira, B.; Anacleto, P.; Vaz-Pires, P.; & Nunes, M.L. Effect of season on the chemical composition and nutritional quality of the edible crab Cancer pagurus. J. Agric. Food Chem., 2009, 57(22), 10814-10824.

Czerniejewski, P.; Bienkiewicz, G.; & Tokarczyk, G. Nutritional quality and fatty acids composition of invasive Chinese mitten crab from Odra Estuary (Baltic Basin). Foods (Basel, Switzerland), 2023, 12(16), 3088.

  • Need for transparent discussion on limitations or assumptions in statistical analyses.

Generally, chemical analysis and color scoring are two mainly means to characterize the degree of pigmentation of crustaceans. Currently, color scoring is widely applicable for color assessment of aquatic animals such as crabs and shrimps. In the present study, the hepatopancreas redness crabs were distinguished by the method of color scoring through perceived spectrophotometer (a*). The color scoring is achieved by perceived visually or spectrophotometer, which results in low accuracy due to perceived color may change depending on the amount of water and environment. We have clarified this perspective in the Discussion section of revised manuscript.

  • Limited exploration of practical applications or implications for the seafood industry, consumers, or resource management.

Our result shown that The redder hepatopancreas, the higher the nutritional quality of male crabs. The local farmers could consider increasing their income through harvesting male crabs with redder hepatopancreas by adjusting the diets and environmental condition (Wu et al., 2011; Jie et al., 2017; Jiang et al., 2022). In crabs, the desired coloration can be achieved by including in the feed astaxanthin, which is the carotenoid responsible for the natural colour (Zhang et al., 2024). However, the cost of synthetic astaxanthin is high, which adds significantly to the costs of feed and production. Fortunately, feeding β-carotene or products rich in β-carotene can provide an alternative and cheaper means to achieve the desired coloration due to crustaceans can convert different dietary carotenoids (including canthaxanthin, lutein or zeaxanthin) into astaxanthin (Long et al., 2017, Wu et al., 2017). We have clarified this perspective in the Discussion section of revised manuscript.

Reference:

Jie, H.; Xuan, F.; Shi, H.; Xie, J.; Wei, W.; Gengshen, W.; & Wenjun, X. Comparison of nutritional quality of three edible tissues of the wild-caught and pond-reared swimming crab (Portunus trituberculatus) females. Lwt-Food Sci Technol, 2017, 75, 624-630.

Jiang, X.; Xie, Z.; Wade, N.M.; Truong, H.H.; Yang, Y.; & Wu, X. Using response surfaces to explore the interactive effect of dietary astaxanthin and β-carotene on growth and antioxidant capability of juvenile Chinese mitten crab, Eriocheir sinensis. Aquaculture, 2022.

Wu, X.; Wang, Z.; Cheng, Y.; Zeng, C.; Yang, X.; & Lu, J. Effects of dietary phospholipids and highly unsaturated fatty acids on the precocity, survival, growth and hepatic lipid composition of juvenile Chinese mitten crab, Eriocheir sinensis (H. Milne‐Edwards). Aquac Res, 2011, 42, 457-468.

Zhang, L., Zhang, R., Jiang, X., Wu, X., & Wang, X. (2024). Dietary supplementation with synthetic astaxanthin and DHA interactively regulates physiological metabolism to improve the color and odor quality of ovaries in adult female Eriocheir sinensis. Food chemistry, 430, 137020.

Long, X., Wu, X., Zhao, L., Liu, J., Cheng, Y., 2017. Effects of dietary supplementation with Haematococcus pluvialis cell powder on coloration, ovarian development and antioxidationcapacity of adult female Chinese mitten crab, Eriocheir sinensis. Aquaculture. 473, 545-553.

Wu, X., Zhao, L., Long, X., Liu, J., Su, F., Cheng, Y., 2017. Effects of dietary supplementation of Haematococcus pluvialis powder on gonadal development, coloration and antioxidant capacity of adult male Chinese mitten crab (Eriocheir sinensis). Aquaculture Research. 48, 5214-5223.

Reviewer 2 Report (New Reviewer)

Comments and Suggestions for Authors

Abstract:

·         The abstract clearly states the study's objective, methods, main findings, and implications. However, it could benefit from a more concise presentation of the primary outcome to enhance its focus.

·         The abstract effectively highlights the significant findings, such as the association between hepatopancreas redness and the contents of specific carotenoids, amino acids, and fatty acids in both male and female crabs. However, it might improve reader comprehension by briefly summarizing the most impactful results or emphasizing the findings directly related to the study's primary hypothesis.

·         The mention of statistical significance (P < 0.05) is crucial for demonstrating the reliability of the findings. However, the abstract could benefit from specifying the statistical analyses used, as this would provide insight into the study's methodological strength.

1. Introduction

·         It could further benefit from a brief overview of this species's global distribution and economic significance to contextualize its importance beyond nutritional aspects.

·         The review of existing studies on the nutritional variability in Eriocheir sinensis is comprehensive. Yet, it could be enhanced by discussing more recent research that highlights the current gaps in knowledge or conflicting findings, providing a stronger rationale for the current study.

·         While the introduction outlines the general aim of the study, it could articulate more clearly the specific research gaps the study intends to fill. A more detailed justification for focusing on hepatopancreas redness as an indicator of nutritional quality would strengthen the premise of the research.

·         The introduction mentions the study's objectives but stops short of explicitly stating the hypotheses under investigation. Including clear, specific hypotheses could sharpen the focus and guide readers on the expected outcomes of the study.

·         The introduction uses specialized terms related to the study subject. While generally appropriate, it would be helpful for readers if critical terms (e.g., "hepatopancreas redness") were defined more explicitly, including why this particular trait is of interest.

·         The flow from general background to specific study aims is logical, but transitions between topics could be smoother to enhance readability. A tighter narrative that directly connects the species' significance to the study's focus would be beneficial.

2. Materials and Methods

·         The methodology for collecting and preparing Eriocheir sinensis samples is clearly described. However, it might be beneficial to include information on any criteria used for selecting the pond for sampling to ensure the representativeness of the sample population.

·         The division of crabs into light and dark color groups based on hepatopancreas values is well-explained. However, further justification on why the top and bottom 20% were chosen for this classification would strengthen the rationale behind these groupings.

·         The techniques for measuring color parameters and conducting paraffin section analysis are briefly mentioned, referencing detailed protocols in supporting information. Including a summary of key steps or considerations directly in the manuscript could improve its standalone comprehensiveness.

·         The manuscript outlines the procedures for analyzing total sugar, lipids, protein, carotenoids, amino acids, and fatty acids, referencing specific methods and standards. It would be helpful to discuss any modifications made to these standard methods and the rationale behind these choices.

3. Results

·         Ensure consistency in the presentation of data across tables and figures. Any discrepancies or unexpected findings should be addressed directly in the text to guide the reader's understanding.

4. Discussion

·         The discussion effectively highlights the positive correlation between hepatopancreas chroma and nutritional quality, emphasizing the role of carotenoids and fatty acids. However, it could further elaborate on the mechanistic underpinnings of these correlations. For instance, discussing how carotenoids influence the nutritional quality beyond serving as vitamin A precursors could provide a more nuanced understanding.

·         The gender-specific differences in nutritional composition related to hepatopancreas chroma are intriguing but warrant a more detailed explanation. The manuscript could benefit from a discussion on the physiological or evolutionary reasons behind these differences, providing a deeper insight into the biological roles of these nutrients in crab development and reproduction.

·         While the manuscript references previous studies on crustacean nutrition and chroma, it could more robustly contextualize its findings within the broader literature. Specifically, comparing and contrasting the results with similar studies in other crustacean species would strengthen the argument and highlight the study's contribution to the field.

·         The discussion on consumer preference based on hepatopancreas chroma and its link to nutritional quality is valuable. However, incorporating consumer studies or market analysis data could solidify consumer behavior and preference claims.

·         The study's methodology, particularly assessing hepatopancreas chroma and its correlation with nutritional parameters, is crucial to the discussion. A more detailed critique of the methods used, including any limitations or potential biases in color measurement and nutritional analysis, would enhance the manuscript's rigor.

·         The manuscript briefly mentions the need for further research into the genetic components influencing crab nutrition and the environmental and dietary factors affecting crab chroma. Expanding on these suggestions by proposing specific study designs or research questions could provide a clearer direction for future work.

·         The conclusion briefly summarizes the study's main findings but could further explore the practical implications of this research. For instance, discussing potential strategies for crab cultivation and selection to enhance nutritional quality based on hepatopancreas chroma would be valuable for producers and consumers.

Comments on the Quality of English Language

The manuscript is well-composed, clearly articulating research findings and discussions. However, there might be occasional grammatical errors or areas where phrasing could be optimized for clarity. These issues do not significantly impede comprehension, but addressing them would enhance the overall readability and professionalism of the document.

Round 2

Reviewer 1 Report (Previous Reviewer 1)

Comments and Suggestions for Authors

Accept after minor revision 

Comments on the Quality of English Language

Minor editing of English language required

This manuscript is a resubmission of an earlier submission. The following is a list of the peer review reports and author responses from that submission.

Round 1

Reviewer 1 Report

Comments and Suggestions for Authors

The abstract lacks a clear and concise statement of the research objective. It should explicitly state the aim and purpose of the study. Consider breaking down the abstract into distinct sections, such as Background, Methods, Results, and Conclusions, to enhance readability. The sentence structure in some parts of the abstract is complex, making it challenging to follow. Simplify sentences for clarity. Proofread for grammar and syntax errors to improve overall writing quality. Some information is repeated in different sections. Ensure that each sentence contributes unique information, avoiding unnecessary repetition. Clarify scientific terminology. For example, explain abbreviations such as "PUFA," "EAA," and "GSI" upon first use to aid readers who may not be familiar with these terms. Include transition sentences to smoothly guide the reader through different study aspects, such as moving from methods to results. Ensure that the citation format is consistent throughout the document. Check if there are any missing or incomplete citations. The abstract lacks a clear concluding statement summarizing the main implications of the study and its potential contributions to the field. The abstract is relatively lengthy. Aim for brevity while ensuring that all essential information is conveyed effectively. Consider adding keywords that reflect the main themes of the research, facilitating searchability and indexing. If available, consider incorporating visual elements such as graphs or charts to illustrate key relationships mentioned in the abstract.

In the results section, the text mentions several statistical results but lacks clarity on the significance levels and p-values. It is important to provide precise statistical information, and if there are no significant differences, this should be explicitly stated. The repetition of information about the differences or lack thereof between light and dark color groups in different parts of the text might be condensed for clarity and brevity. The study reports numerous statistical differences in various parameters, but it's crucial to discuss the biological significance of these findings. Why are these differences important, and how do they contribute to the overall understanding of the species? While there are textual descriptions, visual aids like tables, graphs, or charts could enhance the presentation of the data, making it easier for readers to grasp the information. The section lacks a clear interpretation of the results and how they align with the study's objectives. A conclusion that synthesizes the findings and their implications for the broader context should be included.

The discussion mentions that no previous studies are reporting the relationship between hepatopancreas redness and the qualities of Eriocheir sinensis. However, it would be beneficial to acknowledge any existing gaps in the literature or potential limitations in the current understanding of this relationship. The study primarily focuses on Eriocheir sinensis, and it would be useful to discuss the potential limitations of generalizing the findings to other crab species or even other crustaceans. Acknowledging the species-specific nature of the study's conclusions can enhance the overall robustness of the research. The discussion mentions opposing trends observed in the levels of certain compounds, such as carotenoids, amino acids, and fatty acids, between males and females. Providing potential explanations or hypotheses for these gender-specific variations would strengthen the discussion and help readers better understand the complexities of the findings. While the study establishes correlations between hepatopancreas redness and various qualities, it is crucial to emphasize that correlation does not imply causation. Discussing potential confounding factors or alternative explanations for the observed relationships can add nuance to the interpretation of results. The conclusion could underscore the importance of additional research to validate and expand upon the current findings. Proposing specific avenues for future investigations would contribute to the ongoing scientific discourse on the topic. If there were any limitations or assumptions in the statistical analyses performed, these should be transparently discussed. Addressing potential statistical challenges or limitations can enhance the credibility of the study. While the study discusses the nutritional quality and flavor implications of hepatopancreas redness, practical applications or implications for the seafood industry, consumers, or resource management could be explored further.

Comments on the Quality of English Language

Moderate editing of English language required

Reviewer 2 Report

Comments and Suggestions for Authors

This article discusses an interesting topic; however, I do not feel that the topic is in line with the requirements of Foods. I believe there are other journals that could fit better the scope of this paper. I would recommend submitting this paper to those journals after extensive English check.

Comments on the Quality of English Language

English should be checked, and typos should be corrected. There are sentences that are difficult to understand through the manuscript starting from the 1st phrase of the abstract.

Another difficult phrase: "Aim to clarify the relationship between hepatopancreas redness (a*) and Eriocheir sinensis of nutritional quality and flavour, the body characteristic, chroma (Cuirass, abdomen, hepatopancreas and gonads), histology, as well as the concentrations of carotenoids, protein, lipid, total sugar, amino and fatty acids were detected between the different color groups."

Reviewer 3 Report

Comments and Suggestions for Authors

This paper investigates how the redness of the hepatopancreas in the Chinese mitten crab relates to its nutritional quality. The study focuses on the Chinese mitten crab, a species native to the East Pacific coast of China and popular for its high protein content and unique flavor and aims to establish a link between the hepatopancreas's redness and the crab's nutritional qualities, such as carotenoid content, essential amino acids, and unsaturated fatty acids. As a result, the study found positive correlations between hepatopancreas redness and certain nutritional components, revealing gender-specific differences. However, the manuscript lacked control experiments and didn’t take some very important factors into account, which may affect the results. It should be reconsidered after major revision.

1. How does the chosen sample size and source reflect the species' broader population?

2. Did you consider the environmental and genetic influence; how might they impact the results?

3. The study does not mention a control group, which is crucial for comparative analysis and validating the results.

4. Did you consider the seasonal variability? Conducting similar studies across different seasons would help in understanding how seasonal changes affect the crabs.

5. Investigating how diet influences hepatopancreas redness and nutritional content could add depth to the findings.

6. A deeper investigation into the physiological mechanisms behind the observed gender-specific differences would be enlightening.

Round 2

Reviewer 1 Report

Comments and Suggestions for Authors
  1. The following point must be addressed before moving to the next stage

  2. Abstract:

    • Lack of a clear and concise statement of the research objective.
    • Complex sentence structure and challenging readability.
    • Grammar and syntax errors affect overall writing quality.
    • Repetition of information in different sections.
    • Inconsistency in citation format and missing/incomplete citations.
    • Absence of a clear concluding statement summarizing the main implications.
    • Relatively lengthy abstract without adequate brevity.
    •  
  3. Results Section:

    • Lack of clarity on significance levels in statistical results.
    • Repetition of information about differences or lack thereof between groups.
    • Need for discussion on the biological significance of reported statistical differences.
    • Absence of a clear interpretation of results aligning with study objectives.
    • Lack of a concluding section that synthesizes findings and discusses broader implications.
    •  
  4. Discussion Section:

    • Failure to acknowledge existing gaps in the literature or potential limitations.
    • Lack of discussion on the species-specific nature of the study's conclusions.
    • Insufficient exploration of potential limitations in generalizing findings to other species.
    • Need for providing explanations or hypotheses for gender-specific variations in compounds.
    • Insufficient emphasis on the correlation-does-not-imply-causation principle.
    • Lack of discussion on potential confounding factors or alternative explanations for observed relationships.
    • Absence of a clear emphasis on the importance of additional research and proposing specific avenues for future investigations.
    • Need for transparent discussion on limitations or assumptions in statistical analyses.
    • Limited exploration of practical applications or implications for the seafood industry, consumers, or resource management.
Comments on the Quality of English Language

Moderate editing of English language required

Author Response

Dear Sir or Madam:

Thank you for your comments concerning our manuscript entitled “Relationships between hepatopancreas redness and Chinese mitten crab (Eriocheir sinensis) of body characteristics, carotenoids composition and nutritional quality” (foods-2776876). Those comments are all valuable and very helpful for revising and improving our paper, as well as the important guiding significance to our researches. We have studied comments carefully and have made correction which we hope meet with approval. The main corrections and responds are as following:

Abstract:

  1. Lack of a clear and concise statement of the research objective.

Thank you. We have rewritten the statement of the research objective according to your suggestion. We hope that our revisions are proper and satisfactory.

  1. Complex sentence structure and challenging readability.

I am sorry for your question. We rewrote the complex sentences in our revised manuscript.

  1. Grammar and syntax errors affect overall writing quality.

According to your suggestion, we have made all the necessary editorial changes throughout the manuscript. With all the changes we have made, we wish that this manuscript has been clarified and strengthened.

  1. Repetition of information in different sections.

In our revised manuscript, the repetitive information was deleted according to your suggestion.

  1. Inconsistency in citation format and missing/incomplete citations.

According to your suggestion, we revised the citation in revised manuscript.

  1. Absence of a clear concluding statement summarizing the main implications.

Tank you four your suggestion. We have added a clear concluding statement in Abstract Section.

  1. Relatively lengthy abstract without adequate brevity.

We revised the Abstract Section according to your suggestion, we wish that our revision is proper and satisfactory.

Results Section:

  1. Lack of clarity on significance levels in statistical results.

Thank you. According to your suggestion, the significant levels of our results were clarified in the revised manuscript.

  1. Repetition of information about differences or lack thereof between groups.

According to your suggestion, the duplicate information about differences were deleted, and the differences between the different groups were clarified and refined in Results Section.

  1. Need for discussion on the biological significance of reported statistical differences.

Thank you. We clarified the biological significance of our statistical differences in our revised manuscript.

  1. Absence of a clear interpretationof results aligning with study objectives.

We provided reasonably interpretations for the significant differences between the different groups according to your suggestion.

  1. Lack of a concluding section that synthesizes findings and discussesbroader implications.

In the Result Section, we provided concluding sections of synthesizes findings and discusses according to your suggestion.

Discussion Section:

  1. Failure to acknowledge existing gaps in the literature or potential limitations.

Thank you. It is well known that the hepatopancreas were the mainly edible parts of Chinese mitten crab with high nutrition. Some previous researches had revealed that the chroma of carapace, hepatopancreas and gonad have significantly relationships to regions, size, ages and culture modes (Wang et al., 2017; Wu et al., 2020; Zhang et al., 2020; Li et al., 2020). In addition, some researches were focused on the relationship of hepatopancreas yellowness to nutritious quality and health in recent years (Wang et al., 2021a,b; ). However, there were limited researches reporting the relationship between the hepatopancreas redness and qualities of E. sinensis. We have provided information of the gaps and limitations in the previous literature in Discussion Section. Therefore, our study investigated the relationship between hepatopancreas a* and quality of crabs. It must be clarified that there might be limitations on our results due to the crabs were sampled from one farming, and they belonged the same breed. The further researches are needed for discussing other crab species or even other crustaceans. We have clarified this perspective in the Discussion section of revised manuscript.

References:

Wang, H.N.; Liu, Q.; Wu X.G.; Jiang, X.D.; Xiao, Q.Z.; & Cheng, Y.X. Comparison of the growth and development of adult Eriocheir sinensis between one year precocious family and two year normally mature family during the second year culture stage. Freshwater Fisheries, 2017, 47(3), 84-89. https://doi.org/10.13721/j.cnki.dsyy.2017.03.013

Wu, X.G.; Zhu, S.C.; Zhang, H.C.; Liu, M.M.; Wu, N.; Pan, J.; Luo, M.; Wang, X.C.; & Cheng, Y.X. Fattening culture improves the gonadal development and nutritional quality of male Chinese mitten crab Eriocheir sinensis. Aquaculture, 2020, 518. https://doi.org/10.1016/j.aquaculture.2019.734865

Zhang, L.; Yin, M.Y.; & Wang, X.C. Meat texture, muscle histochemistry and protein composition of Eriocheir sinensis with different size traits. Food Chemistry, 2020, 338(Feb.15), 127632. https://doi.org/10.1016/j.foodchem.2020.127632

Li, Q.Q.; Zu, L.; Cheng, Y.X.; Wade, N.M.; Liu, J.G.; & Wu, X.G. Carapace color affects carotenoid composition and nutritional quality of the Chinese mitten crab, Eriochier sinensis. Lwt-Food Science And Technology, 2020, 126. https://doi.org/10.1016/j.lwt.2020.109286

Wang, Q.J.; Jiang, X.D.; Yao, Q.; Long, X.W.; Xu, Y.P.; Wu, M.; & Zhang, D.M. Comparative study on the nutrition composition of adult male Chinese mitten crab (Eriocheir sinensis) with different coloured hepatopancreases. Aquaculture Research,  2021a, 52(1), 196-207. https://doi.org/10.1111/are.14881

Wang, Q.J.; Zhang, B.Y.; Jiang, X.D.; Long, X.W.; Zhu, W.L.; Xu, Y.P.; Wu, M.; & Zhang, D.M. Comparison on nutritional quality of adult female Chinese mitten crab (Eriocheir sinensis) with different colored hepatopancreases. Journal of Food Science, 2021b, 86(5), 2075-2090. https://doi.org/10.1111/1750-3841.15664

  1. Lack of discussion on the species-specificnature of the study's conclusions.

In the present study, our results found positive correlations between hepatopancreas redness and certain nutritional components, revealing gender-specific differences. However, our conclusion might be species-specific owing to different biochemical composition among different crabs. Previous study indicated that the female Eriocheir japonica had the higher levels of PUFA, while E. sinensis had a higher amino acid and EAA contents (Wang et al., 2020). The ratio of PUFA n-3/n-6 in E. sinensis was lower than that in blue crab (Callinectes sapidus) (Celik et al., 2004; Chen et al., 2007). In addition, the significant differences of mineral contents were revealed between Chinese mitten crab and the other crabs, such as green crab (Carcinus maenus), blue crab and swim crab (Portunus pelagicus) (Chen et al., 2007). According to your suggestion, we have added the discussion on the species-specific nature of our present conclusions in the Discussion Resection.

References:

Wang, Z.; Zu, L.; Li, Q.; Jiang, X.; Xu, W.; Soyano, K.; Cheng, Y.; & Wu, X. A comparative evaluation of the nutritional quality of Eriocheir sinensis and Eriocheir japonica (Brachyura, Varunidae). Crustaceana, 2020, 93, 567-585. https://doi.org/10.1163/15685403-bja10024

Chen, D.W.; Zhang, M.; & Shrestha, S. Compositional characteristics and nutritional quality of Chinese mitten crab (Eriocheir sinensis). Food Chemistry, 2007, 103, 1343-1349. https://doi.org/10.1016/j.foodchem.2006.10.047

Celik, M.; Türeli, C.; Çelik, M.; Yanar, Y.; Erdem, Ü.; & Küçükgülmez, A. Fatty acid composition of the blue crab (Callinectes sapidus Rathbun, 1896) in the north eastern Mediterranean. Food Chemistry, 2004, 88, 271-273. https://doi.org/10.1016/j.foodchem.2004.01.038

  1. Insufficient exploration of potential limitations in generalizing findings to other species.

According to your suggestion, we have supplied the information for exploring limitations of our finding to other species in our revised manuscript.

  1. Need for providing explanations or hypotheses for gender-specific variations in compounds.

Thank you for your suggestion. As for the differences on the carotenoids, amino acids and fatty acids between males and females, the potential explanations and hypotheses were supplied in the Discussion section of revised manuscript. Of course, further researches are needed for clarifying the mechanism of differences between the sexes.

  1. Insufficient emphasis on the correlation-does-not-imply-causation principle.

In present study, our results indicated the positive correlations between hepatopancreas redness and certain nutritional components. However, this correlation was not meaning that hepatopancreas redness was the reasons for different nutritional components. Generally, the crabs chroma and biochemical compositions were usually affected by diets, environmental and genetic factors (Wu et al., 2011; Kong et al., 2012; Jie et al., 2017; Jiang et al., 2022). Therefore, additional studies must be conducted to reveal the underlying reasons and molecular mechanism. According to your suggestion, we have added some reasonable inferences for our findings in revised manuscript. In addition, we have emphasized that the future researches are needed for revealing the underlying molecular mechanism in Discussion Resection.  

References:

Jiang, X.; Xie, Z.; Wade, N.M.; Truong, H.H.; Yang, Y.; & Wu, X. Using response surfaces to explore the interactive effect of dietary astaxanthin and β-carotene on growth and antioxidant capability of juvenile Chinese mitten crab, Eriocheir sinensis. Aquaculture, 2022. https://doi.org/10.1016/j.aquaculture.2022.738196

Kong, L.; Cai, C.; Ye, Y.; Chen, D.; Wu, P.; Li, E.; Chen, L.; & Song, L. Comparison of non-volatile compounds and sensory characteristics of Chinese mitten crabs (Eriocheir sinensis) reared in lakes and ponds: Potential environmental factors. Aquaculture, 2012, 364, 96-102. https://doi.org/10.1016/j.aquaculture.2012.08.008.

Jie, H.; Xuan, F.; Shi, H.; Xie, J.; Wei, W.; Gengshen, W.; & Wenjun, X. Comparison of nutritional quality of three edible tissues of the wild-caught and pond-reared swimming crab (Portunus trituberculatus) females. Lwt - Food Science and Technology, 2017, 75, 624-630. https://doi.org/10.1016/j.lwt.2016.10.014

Wu, X.; Wang, Z.; Cheng, Y.; Zeng, C.; Yang, X.; & Lu, J. Effects of dietary phospholipids and highly unsaturated fatty acids on the precocity, survival, growth and hepatic lipid composition of juvenile Chinese mitten crab, Eriocheir sinensis (H. Milne‐Edwards). Aquaculture Research, 2011, 42, 457-468. https://doi.org/10.1111/j.1365-2109.2010.02643.x

  1. Lack of discussion on potential confounding factors or alternative explanations for observed relationships.

According to your suggestion, we have supplied some related discussions to clarify our current results in our revised manuscript.

  1. Absence of a clear emphasis on the importance of additional research and proposing specific avenues for future investigations.

Thank you for your suggestion. In our present study the relationships of hepatopancreas redness to Chinese mitten crab of body chroma, carotenoids composition and nutritional quality were revealed. Some reasonable inferences were supplied to clarify the reasons of differences between different groups. Therefore, additional studies must be conducted to confirm our findings. According to your suggestion, we have added the information to emphasize the importance of additional researches. And we also proposed specific avenues of future investigations for clarifying molecular mechanisms, such as transcriptomics and metabolomics studies.

  1. Need for transparent discussion on limitations or assumptions in statistical analyses.

Thank you for your question. The aim of our present study is to clarify the relationship between hepatopancreas chroma and quality of crabs. Therefore, we only detected the the body characteristic, chroma (Cuirass, abdomen, hepatopancreas and gonads), histology, carotenoids, protein, lipid, total sugar, amino and fatty acids were detected between the different red hepatopancreas crabs. Indeed, crabs chroma and biochemical composition were usually affected by diets, environmental and genetic factors (Wu et al., 2011; Kong et al., 2012; Jie et al., 2017; Jiang et al., 2022). In our present study, the crabs were sampled from the same conditions of diet and environment. Therefore, the genetic influence on the carotenoids should be considered primarily in the future. Additionally, our results indicated the positive correlations between hepatopancreas redness and certain nutritional components. However, the underlying reasons and molecular mechanism for our conclusion were not revealed and clarified. We have clarified this perspective in the Discussion section of revised manuscript.

  1. Limited exploration of practical applications or implications for the seafood industry, consumers, or resource management.

In the present study, our results firstly proved that hepatopancreas chroma could be judged through the chroma of cuirass and abdomen in mature crabs, which provided a scientific method for consumers to choose crabs. In addition, our results shown that the redder hepatopancreas crabs had higher levels of carotenoids, which could guide consumers to choose crabs according to their demands for carotenoids. Moreover, the relationship between hepatopancreas chroma and nutritional quality was clarified, which provided the research basis and supports for developing high-quality crab food according to chroma. According to this findings, local farmers could consider increasing their income through harvesting crabs according to hepatopancreas chroma in different sex. Therefore, this work is more meaningful for consumers, and it also provides new insights for the nutritional assessment of the male E. sinensis. We have provided discussions about our results in the practical applications or implications for the seafood industry or consumers in our revised manuscript.

Reviewer 3 Report

Comments and Suggestions for Authors

The revised manuscript is suitable for publication now.

Author Response

Dear Sir or Madam:

Thank you for your comments concerning our manuscript entitled “Relationships between hepatopancreas redness and Chinese mitten crab (Eriocheir sinensis) of body characteristics, carotenoids composition and nutritional quality” (foods-2776876).